# Selection and hybridization shaped the rapid spread of African honey bee ancestry in the Americas

Erin Calfee[1,2]*, Marcelo Nicolás Agra[3], María Alejandra Palacio[3,4], Santiago R. Ramírez[1,2], Graham Coop[1,2]

**1** Center for Population Biology, University of California, Davis, California, United States of America,
**2** Department of Evolution and Ecology, University of California, Davis, California, United States of America,
**3** Instituto Nacional de Tecnología Agropecuaria (INTA), Balcarce, Argentina, **4** Facultad de Ciencias Agrarias, Universidad de Mar del Plata, Balcarce, Argentina

* ecalfee@ucdavis.edu

**Data Availability Statement:** Raw Illumina sequence data generated by this study is available through the NCBI Short Read Archive, PRJNA622776. Access information for all

## Abstract

Recent biological invasions offer 'natural' laboratories to understand the genetics and ecology of adaptation, hybridization, and range limits. One of the most impressive and well-documented biological invasions of the 20th century began in 1957 when *Apis mellifera scutellata* honey bees swarmed out of managed experimental colonies in Brazil. This newly-imported subspecies, native to southern and eastern Africa, both hybridized with and out-competed previously-introduced European honey bee subspecies. Populations of *scutellata*-European hybrid honey bees rapidly expanded and spread across much of the Americas in less than 50 years. We use broad geographic sampling and whole genome sequencing of over 300 bees to map the distribution of *scutellata* ancestry where the northern and southern invasions have presently stalled, forming replicated hybrid zones with European bee populations in California and Argentina. California is much farther from Brazil, yet these hybrid zones occur at very similar latitudes, consistent with the invasion having reached a climate barrier. At these range limits, we observe genome-wide clines for *scutellata* ancestry, and parallel clines for wing length that span hundreds of kilometers, supporting a smooth transition from climates favoring *scutellata*-European hybrid bees to climates where they cannot survive winter. We find no large effect loci maintaining exceptionally steep ancestry transitions. Instead, we find most individual loci have concordant ancestry clines across South America, with a build-up of somewhat steeper clines in regions of the genome with low recombination rates, consistent with many loci of small effect contributing to climate-associated fitness trade-offs. Additionally, we find no substantial reductions in genetic diversity associated with rapid expansions nor complete dropout of *scutellata* ancestry at any individual loci on either continent, which suggests that the competitive fitness advantage of *scutellata* ancestry at lower latitudes has a polygenic basis and that *scutellata*-European hybrid bees maintained large population sizes during their invasion. To test for parallel selection across continents, we develop a null model that accounts for drift in ancestry frequencies during the rapid expansion. We identify several peaks within a larger genomic region where selection has pushed

previously published genomic resources used in this study: HAv3.1 reference genome and gene annotations (NCBI PRJNA471592), recombination map (available by request from Jones et al.), bee genomes from California (NCBI PRJNA385500) and A, C, and M reference populations (NCBI PRJNA216922 and PRJNA294105). Bee metadata, including GPS locations and measured wing lengths are included in S1 Table. Wing images generated by this study are available through Data Dryad: https://doi.org/10.25338/B8T032. Wing images for museum samples of A, C, and M bees are available by request from the Morphometric Bee Data Bank, Institut für Bienenkunde, Oberursel, Germany (https://de.institut-fuer-bienenkunde.de). All climate data was downloaded from WorldClim. org. Scripts are available at https://github.com/ecalfee/bees.

**Funding:** This work was funded by the National Institute of General Medical Sciences of the National Institutes of Health, www.nigms.nih.gov (NIH R01 GM108779 and R35 GM136290, awarded to GC), the Division of Integrative Organismal Systems from the National Science Foundation, www.nsf.gov (NSF No. 1546719, awarded to GC), the North American Pollinator Protection Campaign and Pollinator Partnership, www.pollinator.org/nappc (NAPPC Honey Bee Health Grant, awarded to EC and SR), and the Center for Population Biology UC Davis, https://cpb.ucdavis.edu (Pengelley Award, awarded to EC). The funders had no role in the study design, data collection and analysis, decision to publish, or preparation of the manuscript.

**Competing interests:** The authors have declared that no competing interests exist.

*scutellata* ancestry to high frequency hundreds of kilometers past the present cline centers in both North and South America and that may underlie high-fitness traits driving the invasion.

## Author summary

Crop pollination around the world relies on native and introduced honey bee populations, which vary in their behaviors and climatic ranges. *Scutellata*-European hybrid honey bees (also known as 'Africanized' honey bees) have been some of the most ecologically successful; originating in a 1950s experimental breeding program in Brazil, they rapidly came to dominate across most of the Americas. As a recent genetic mixture of multiple imported *Apis mellifera* subspecies, *scutellata*-European hybrid honey bees have a patchwork of ancestry across their genomes, which we leverage to identify loci with an excess of *scutellata* or European ancestry due to selection. We additionally use the natural replication in this invasion to compare outcomes between North and South America (California and Argentina). We identify several genomic regions with exceptionally high *scutellata* ancestry across continents and that may underlie favored *scutellata*-European hybrid honey bee traits (e.g. *Varroa* mite resistance). We find evidence that a climatic barrier has dramatically slowed the invasion at similar latitudes on both continents. At the current range limits, *scutellata* ancestry decreases over hundreds of kilometers, creating many bee populations with intermediate *scutellata* ancestry proportions that can be used to map the genetic basis of segregating traits (here, wing length) and call into question the biological basis for binary 'Africanized' vs. European bee classifications.

## Introduction

Diverging lineages often spread back into secondary contact before reproductive isolation is complete, and so can hybridize. In hybrid zones, multiple generations of admixture and backcrossing create a natural experiment in which genetic variation is 'tested' in novel ecological and genomic contexts. The mosaic of ancestries in hybrid zones has allowed researchers to uncover the genetic loci associated with reproductive barriers (e.g. [1–3]) and to identify rapidly introgressing high-fitness alleles (e.g. [4–7]). One promising way forward is to compare ancestry patterns across multiple young hybrid zones and test how repeatable the outcome of hybridization is across these evolutionary replicates.

In this study, we use this powerful comparative framework to better understand the genomic basis of fitness and range limits of *scutellata*-European hybrid honey bees, with replicate routes of invasion into North and South America. The range of the western honey bee (*Apis mellifera*) has expanded from Africa, Europe, and western Asia [8] across much of the globe, assisted by colonialism and the ecological diversity of honey bee subspecies [9]. While the Americas have many species of native bees and a long cultural history of beekeeping with honey-producing stingless bees (Meliponini), colonists as early as the 1600s imported European honey bee subspecies for their own apiculture and agriculture uses [10], setting off the first honey bee invasion of the Americas [11]. Through a combination of human-assisted migration and swarming, European honey bees spread across the continent and founded feral populations [10]. Then in 1957, swarms from a newly-imported honey bee subspecies from southern and eastern Africa, *Apis mellifera scutellata*, escaped from an experimental breeding program in Brazil and rapidly dispersed. Widely successful, *scutellata* honey bees both outcompeted and hybridized with European-ancestry populations, creating a rapidly advancing

*scutellata*-European admixed population that expanded north and south across the Americas at 300-500 km/year [12].

Colonies of *scutellata*-European hybrids are likely to respond more strongly to disturbances than colonies with European ancestry (measured as number of stings per minute, reduced time to sting, and longer pursuit distances [13–15]). The spread of these more defensive bees (sensationalized as 'killer bees', see critiques [16–19]) have created new challenges for beekeepers and public health [16, 20]

Control efforts have been largely unsuccessful in slowing the invasion or preventing the spread of *scutellata* ancestry into commercial colonies [12, 16]. However, even without intervention, *scutellata* ancestry is unlikely to outcompete European ancestry in the coldest regions of the Americas because *scutellata*-European hybrid honey bees from the neotropics have low overwinter survival in climates where European bees thrive [21, 22]. Models based on winter temperatures and the physiological cost of thermo-regulation predict northern range limits for the invasion that vary from the Central Californian Coast [23, 24] up to the border with Canada [25]. Thus, the expected impact of *scutellata* ancestry on agriculture and queen bee production in the United States is still poorly defined. Broad surveys show that *scutellata* -like mtDNA and phenotypes are common in northern Argentina and the southern US, and drop off towards more temperate latitudes, indicating that the rapid spread of these traits has dramatically slowed, if not stopped, on both continents [23, 26–31]. However, we lack a genome-wide view of the range limits of *scutellata* ancestry and do not know whether individual high-fitness alleles have already introgressed into higher latitudes.

Previous genomic work on the invasion has shown that *scutellata*-European hybrid honey bees are a genetic mixture of three major genetic groups: A from Africa, C from eastern Europe and M from western Europe [32–37]. Historical sources indicate that the A ancestry is from *A. m. scutellata* [38, 39], while both M and C ancestries are mixtures of multiple subspecies imported from Europe, e.g. *A. m. ligustica* (C), *A. m. carnica* (C), *A. m. mellifera* (M), and *A. m. iberiensis* (M) [10]. Many names have been used previously to refer to *scutellata*-European hybrids in the literature, including 'African honey bees', 'African hybrid honey bees', or 'Africanized honey bees', and the ambiguous acronym 'AHB', with these names being used to describe bees identified as having *scutellata* ancestry on the basis of behavior, morphology, mtDNA, or a range of *scutellata* autosomal ancestry. Given the wide range of *scutellata*-European ancestry that we find in this study, and that *A. m. scutellata* is only one of at least 10 ecologically diverse *Apis mellifera* subspecies native to Africa [38], we will simply use the label *scutellata*-European hybrids for individuals whose autosomal genome is comprised of a mixture of these ancestries.

While the key genes remain unknown, *scutellata*-European hybrid honey bees diverge from European-ancestry bees on a number of traits that may have given them a selective advantage during the invasion: they have higher reproductive rates (including faster development times, proportionally higher investment in drone production and more frequent swarming to found new colonies [12]), they have higher tolerances to several common pesticides [40], and they prove less susceptible to *Varroa* mites, a major parasite [41–45]. Population monitoring studies show that *Varroa* mites are a strong selective force in the wild and that mite infestations in the 1990s likely contributed to the rapid genetic turnover of feral nest sites from European to *scutellata*-European hybrid colonies in Arizona and Texas [28, 29, 36]. European ancestry may have also contributed to the success of the invasion; a recent study of *scutellata*-European hybrid bees in Brazil revealed some European alleles at exceptionally high frequency, but this work was under-powered to detect high-fitness *scutellata* alleles due to elevated genome-wide *scutellata* ancestry (84%) in the Brazilian population [37].

There are also a number of candidate traits that distinguish *scutellata*-European hybrid honey bees from European bees and plausibly contribute to a climate-based range limit for the

invasion. Smaller bodies [46] and higher metabolic rates [25], for example, could give honey bees with high *scutellata* ancestry a competitive advantage in the tropics but come at a cost in cooler climates [24]. In addition to physiological traits, heritable behaviors may also contribute to fitness trade-offs: *scutellata*-European hybrid bees from Venezuela to Arizona preferentially forage for protein-rich pollen (vs. nectar), which supports rapid brood production, but risks insufficient honey stores to thermo-regulate over winter [24, 47, 48].

Other traits associated with *scutellata* ancestry are of central importance to beekeepers, but their role in the invasion is less clear. Stronger colony-defense behaviors have been reported across much of the range of *scutellata*-European hybrid honey bees [16, 39] (with some local exceptions, see [49, 50]). The fitness consequence of these behaviors will depend on the costs of both predation and defense. Similarly, more frequent absconding (leaving a nest site to find another) is undesirable in managed apiaries, but may be adaptive in some environments, e.g. to escape predators or local resource shortages [51]. Selection for these traits is likely to vary across the range of *scutellata*-European hybrid honey bees, depending on the natural and human-mediated environment.

Here we conduct the first comparative study of the *scutellata*-European hybrid honey bee invasion in North and South America. First, we use broad geographic sampling and whole genome sequencing to map the present-day ancestry clines on both continents, and assess the evidence for a climatic barrier preventing the further spread of *scutellata* ancestry. Next, we use genetic diversity within *scutellata* ancestry to study the shared bottleneck within and amongst populations due to the rapid expansion during the invasion. Finally, we develop a null model that includes recent drift and use this model to test for outlier loci that may underlie high-fitness *scutellata*-European hybrid honey bee traits and climatic barriers.

## Results

To survey the current geographic distribution of *scutellata* ancestry in the Americas, we sampled and sequenced freely foraging honey bees across two latitudinal transects, one in California and one in Argentina, formed from the northern and southern routes of invasion out of Brazil (Fig 1). We generated individual low-coverage whole-genome sequence data for 278 bees, and added to this data set 35 recently published high-coverage bee genomes from 6 additional California populations sampled 3-4 years prior [34]. We inferred genome-wide ancestry proportions for each individual using NGSAdmix [52] assuming a model of 3 mixing populations, which clearly map to the *scutellata* (A), eastern European (C), and western European (M) reference panels (Fig 1 and S2 Fig). We leveraged the fact that admixed *scutellata*-European honey bee populations were formed through a recent mixture of known genetic groups to infer the mosaic of A, C and M ancestry tracts across the genome of each bee. For each population, we applied a hidden Markov model that jointly infers the maximum likelihood single-pulse approximation for the generations since mixture and posterior probabilities for local ancestry state, based on read counts from low-coverage sequence data (ancestry_hmm [53]). The average local ancestry estimates within individuals agree closely with the NGSAdmix genome-wide ancestry estimates (S3 Fig, Pearson's $r \geq 0.985$), with the HMM estimating slightly higher minor ancestry for low admixture proportions, likely as a result of some miscalled blocks. Time estimates vary by population, with a median of 47.6 generations in the 62 years since the initial dispersal of *scutellata* queen bees out of São Paulo (see S5 Fig for all time estimates). In this section, we first focus on the distribution of genome-wide 'global' ancestry patterns across the two clines, which we will later compare to the variation in local ancestry at individual loci.

We observe wide hybrid zones mirrored in North and South America. In Argentina, we find the cline in ancestry spans nearly 900km, from 77% *scutellata* (A) ancestry in the north to

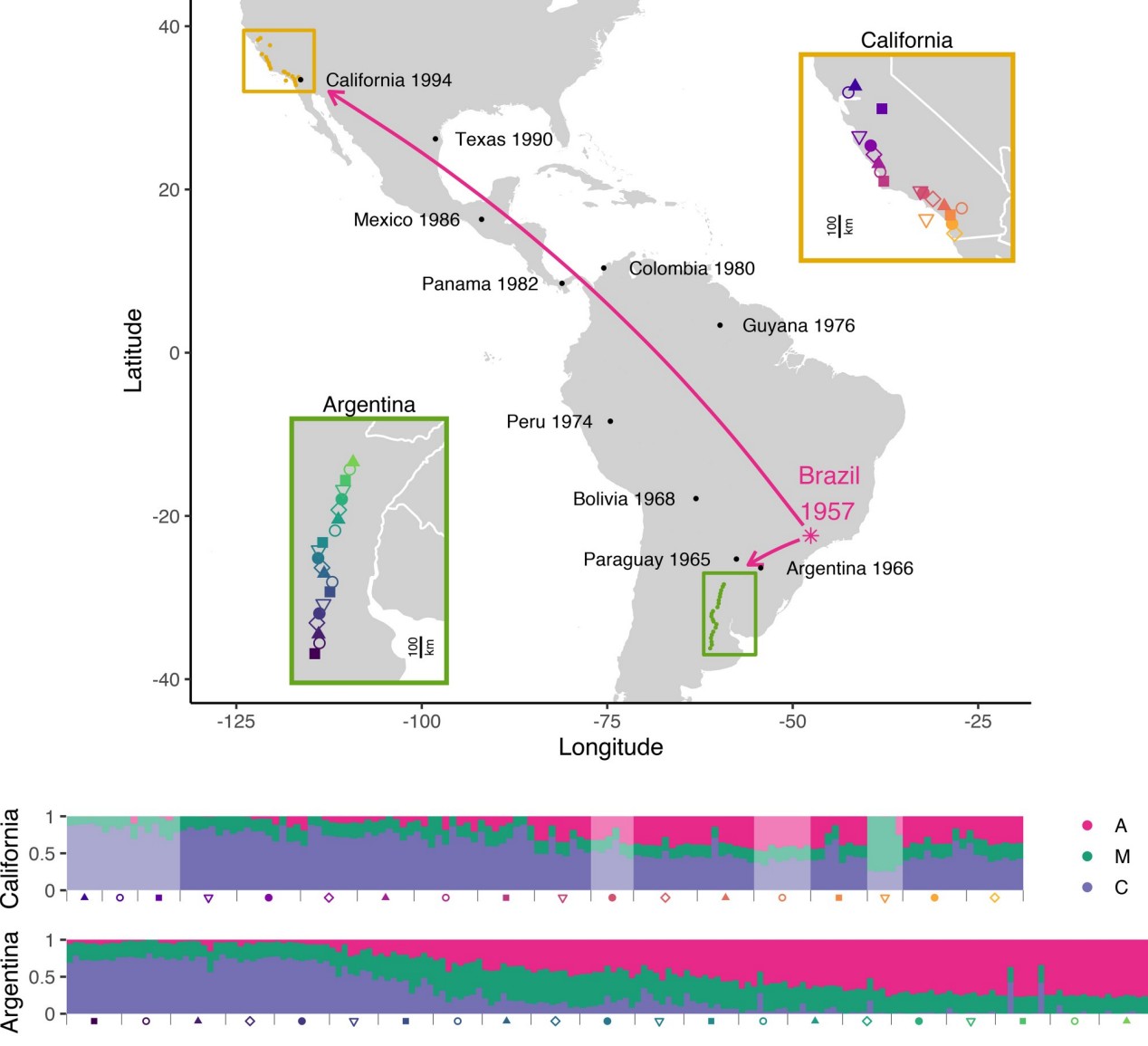

**Fig 1. Spread of *scutellata* ancestry in the Americas.** Map of hybrid zones in California and Argentina, with cartoon arrows depicting the two routes of *scutellata*-European hybrid honey bee invasion out of Rio Claro, São Paulo, Brazil. Dates of first occurrence along the routes of invasion are from [12] [54] and [55], with approximate GPS locations extracted from google maps. Insets zoom in on each hybrid zone to show the mean GPS coordinates for each sampled population. Sampling spanned 646km in California and 878km in Argentina in the north-south direction. Genome-wide *scutellata* (A), eastern European (C), and western European (M) ancestry inferred using NGSAdmix for each bee are shown in a bar chart at the bottom, where each vertical bar is one bee and colors indicate proportion ancestry. Populations are arranged by latitude, with samples closest to Brazil on the right. Light fading indicates that a bee comes from the previously published California data set [34] and was collected in the field 3-4 years prior to the bees from this study. These earlier California samples include one island population, Avalon (Catalina Island), indicated by a yellow triangle. Bees from Avalon have majority M ancestry, in contrast to all mainland California bees which have predominantly A and C ancestry. The underlying maps were created by plotting geographic data from the CIA World DataBank II [56] in R [57] using ggplot [58].

less than 5% to the south in Buenos Aires Province. The current geographic range of A ancestry in South America is broadly consistent with prior studies using a smaller number of genetic markers (e.g. [26, 27, 35, 59]), though the geographic and genetic resolution of these studies is too limited for detailed comparison. In North America, we find that honey bees in California have up to 42% A ancestry in the south, tapering down to approximately 0% in Davis, our northernmost sampling site. In comparison, earlier extrapolations based on mitochondrial

surveys may have somewhat overestimated genome-wide A ancestry in California (e.g. 65% of foraging bees in San Diego County [30] and 17% in Monterey County [31] carry A mtDNA haplotypes). We also find excess A-like mtDNA diversity in California. While this finding is potentially consistent with *scutellata* maternal lines being favored during the expansion into Southern California, this pattern is not strongly replicated in South America and even in North America, A mitochondria do not appear to have introgressed far past the northern range limit for nuclear A ancestry (S26 Fig).

Alongside our genomic cline, we find a corresponding phenotypic cline in worker fore wing size: closer to the equator, sampled bees have increasing A ancestry and shorter wings (Fig 2). By fitting a linear model to predict wing length from genome-wide ancestry, we find that A ancestry can explain a difference of -0.72mm, approximately an 8% reduction in wing length ($P = 3.65 \times 10^{-23}$, $R^2 = 0.31$, $n = 269$; see S7 Fig). We tested for a main effect and an interaction term for the South American continent, and found no significant differences in wing length ($P = 0.81$) or its association with ancestry ($P = 0.86$) between the two clines. Thus, in contrast to the rise of dispersal-enhancing traits in other recent invasions (e.g. [60–63]), we see no evidence of a bias for longer wings at larger dispersal distances (California). Genetic crosses have shown that wing length differences between ancestries have a genetic basis [15] and the wing length patterns we observe here are consistent with expectations of an additive polygenic cline based on genome-wide ancestry alone (Fig 2). However, these phenotypic clines could alternatively be caused purely by developmental plasticity or sorting of within-ancestry genetic variation along a latitudinal gradient. Preliminary evidence that other factors may contribute to the wing length clines observed here comes from a 1991 survey showing that wing length was positively correlated with latitude in California's feral bee populations before the reported arrival of *scutellata*-European hybrid honey bees [64]. From field-based sampling alone, it remains unclear what portion of the observed phenotypic clines are ancestry-driven. We performed admixture mapping to test for genetic loci underlying ancestry-associated differences in wing length and did not identify any loci meeting genome-wide significance (S9 Fig).

Our genomic results indicate that the geographic distribution of *scutellata* ancestry is presently constrained by climatic barriers, not dispersal. Historical records document an initial rapid spread of *scutellata*-European hybrid honey bees from their point of origin in Rio Claro, São Paulo, Brazil, followed by the formation of seemingly stable hybrid zones at similar latitudes in North and South America. Yet to reach this same latitude, northern-spreading bees had to travel more than five times the distance as southern-spreading *scutellata*-European hybrid honey bee populations.

To more precisely infer the current shape and position of the two hybrid zones, we fit a classic logistic cline model to inferred genome-wide individual *scutellata* ancestry proportions [65–67]:

$$A_i = \frac{M}{1 + e^{-b(x_i - c)}}, \tag{1}$$

where $A_i$ is the genome-wide *scutellata* ancestry proportion inferred for the $i^{th}$ individual bee, $x_i$ is their latitude, $M$ is the asymptotic maximum *scutellata* ancestry approaching the equator, which is set at 0.84 (i.e. frequency in Brazil [37]), $c$ is the cline center, and $w = |^4/_b|$ is the cline width (i.e. the inverse of the steepest gradient at the center of the cline).

Each degree latitude corresponds to approximately 111km and presents a natural way to compare cline position and shape between the two zones. We fit this model in R using non-linear least squares (although maximum likelihood or Bayesian estimation are generally preferred when the errors can be fully parameterized, here least squares allows for unknown drift variance in addition to binomial sampling variance). We find that the two hybrid zones have

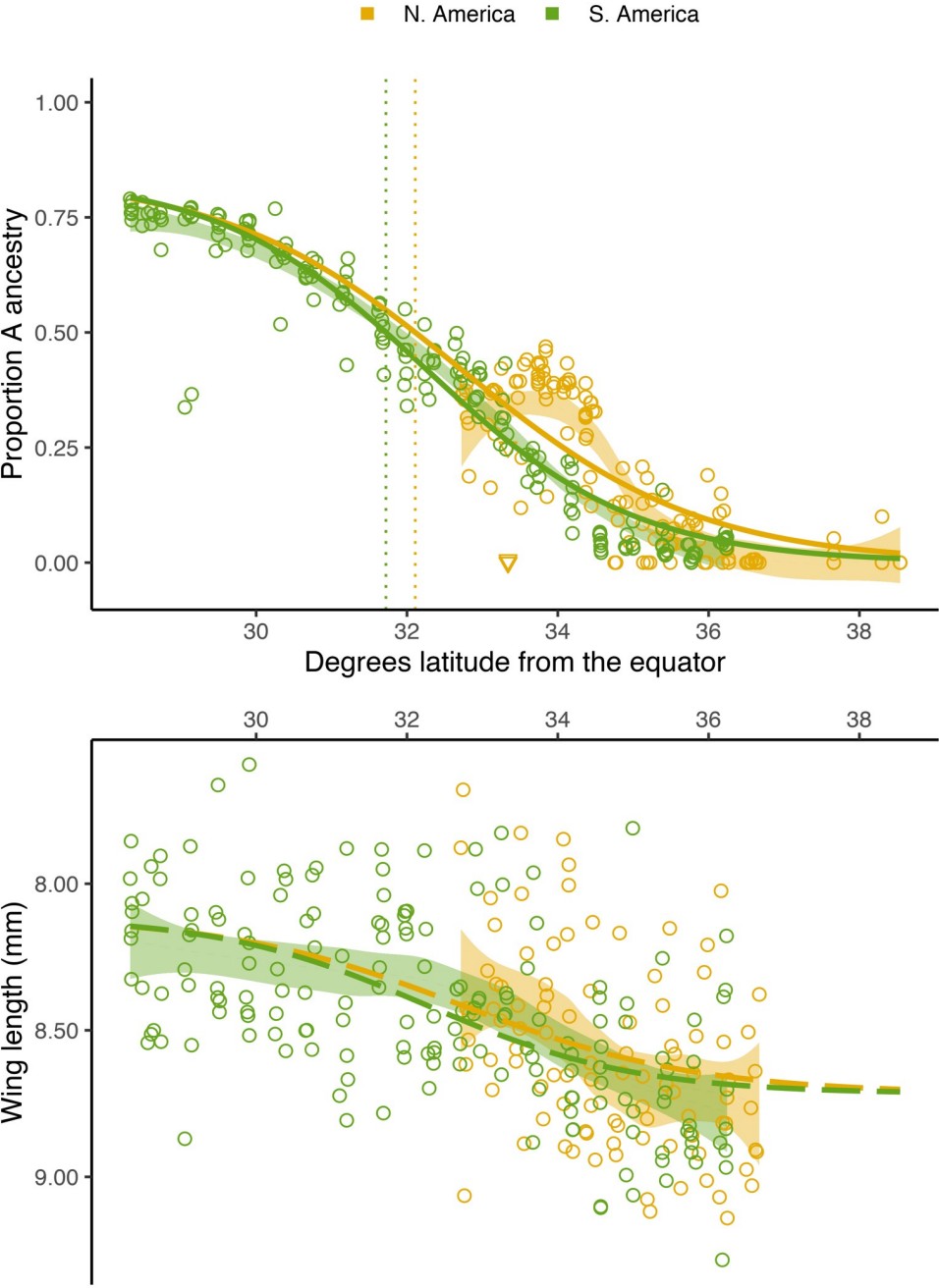

**Fig 2. Clines across latitude.** Genome-wide ancestry estimates (top) and fore wing lengths (bottom) for individual bees, plotted across absolute latitude and colored by continent. Shading indicates the 95% confidence intervals for loess curves of the raw data. We also overlay several model-fitted clines: In the top panel, solid curves show the North and South American logistic cline fits for ancestry predicted by latitude, with dotted vertical lines marking the latitude at which bees have predicted 50% *scutellata* (A) ancestry, based on these curves. Samples from Avalon are displayed as orange triangles; Catalina Island has a distinct ancestry composition from mainland California populations and low A ancestry for its latitude. In the bottom panel, dashed curves show the expected phenotypic cline if wing lengths were fully determined by the clines in ancestry depicted in the top panel. To get these predicted wing lengths, we used the mean ancestry cline as input to the best-fit linear model between ancestry and wing length. Note that the y-axis for wing lengths is reversed (smaller wings are higher) to simplify visual comparisons between the top and bottom panels.

strikingly similar positions (Fig 2), with cline centers that differ by less than half a degree (32.72˚N vs. 32.26˚S), and no statistically significant difference in cline steepness. To better understand the mechanisms underlying this parallelism between continents, we tested four possible explanatory climate variables to see if we could identify a better predictor for *scutellata* ancestry across our two zones than latitude: Mean annual temperature (˚C), mean temperature of the coldest quarter (˚C), minimum temperature of the coldest month (˚C), and mean annual precipitation (cm) (downloaded from WorldClim.org [68]). We fit clines for both hybrid zones jointly using these four environmental variables in turn as predictors in Eq 1 in place of $x_i$, and compared these results to fits based on absolute latitude and, as a neutral dispersal model, distance from São Paulo.

We find that latitude is the best individual predictor of genome-wide global ancestry, and mean annual temperature the second-best predictor, as assessed by AIC (see S2 Table). While latitude provides the best-fitting cline, we find it unlikely that latitude or daylight per se is the relevant selection gradient. Temperature and precipitation are closely coupled to latitude across our transect in Argentina, so nearly all of our resolution to disentangle latitude from environmental gradients comes from micro-climates within California, and for precipitation, the contrast between continents (S6 Fig). However, we failed to identify specific environmental variables that may be driving the relationship with latitude, either because we did not include the relevant environmental variable(s) or because the climate data does not reflect the selection environment of sampled bees, e.g. due to mismatches in scale or selective habitat use by bees within a foraging range.

Despite limited resolution on the climate variables driving the latitudinal gradient, our comparative framework allows us to firmly reject a neutral model based on distance from the point of introduction in Brazil, because a single dispersal rate cannot generate predictions that simultaneously fit the clines in North and South America well (see S2 Table).

In addition to these global ancestry estimates, we measure variation in local ancestry frequencies across the genome, which are informative about recent evolutionary history. *Scutellata* ancestry frequencies at individual loci will vary around their genome-wide mean due to finite sampling, but also evolutionary processes, including drift and selection. If two populations have shared gene flow post-admixture, at loci where one population has higher than average *scutellata* ancestry frequencies, the second population will also tend to have higher than average *scutellata* ancestry. We capture this genetic signature in an ancestry covariance matrix, where each entry represents how much a pair of populations co-deviate in locus-specific *scutellata* ancestry away from their individual genome-wide means (Fig 3). We expect ancestry covariances to build up along each route of the *scutellata*-European hybrid honey bee invasion as a result of shared drift post-admixture. Indeed, we do observe positive ancestry covariances for nearby populations within each hybrid zone. We attribute this pattern to shared demographic history, but also note that weak selection for a specific ancestry at many loci genome-wide could also generate these positive covariances. Unexpectedly, we find that populations in more temperate North and South America, i.e. at opposite ends of the expansion, have higher ancestry correlations with each other than with populations situated between them. This robust signal is a general pattern that holds true on average across chromosomes (S14 Fig), and so isn't driven by individual outlier loci, and persists across recombination rate bins (S15 Fig). These similar ancestry patterns in geographically distant populations are potentially consistent with a genome-wide signature of convergent selection to cooler climates or convergent selection by beekeepers at higher latitudes. Another possible explanation is recent long-distance migration (e.g. international bee exports); however, we investigated genetic covariance patterns within A, C, and M ancestries and found no clear evidence of gene flow between the high-latitude cline endpoints (see Methods).

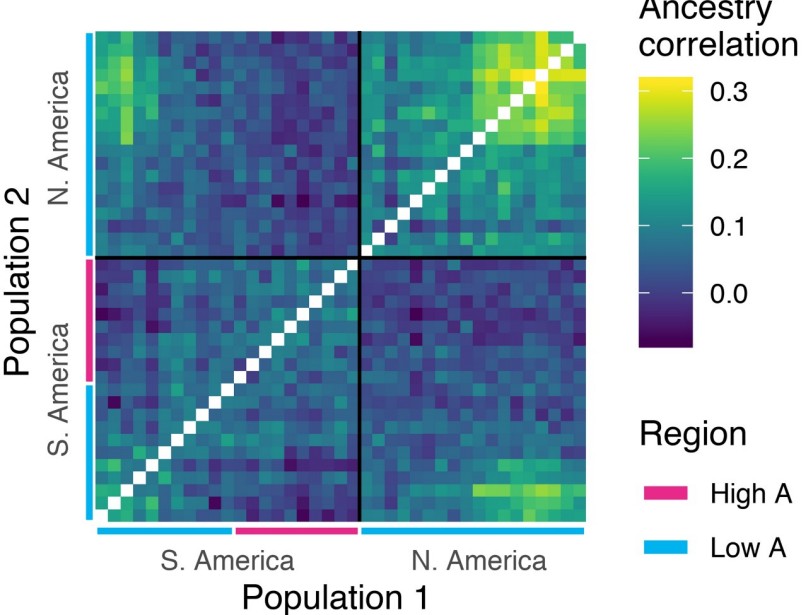

**Fig 3. Correlated ancestry across populations.** Shared drift in ancestry shown as an ancestry correlation matrix (see Methods). Populations are ordered by latitude and diagonals are left blank (within-population correlations = 1). Low and high A ancestry regions of each hybrid zone are defined relative to the estimated latitude of the cline center for genomewide ancestry. About half of the sampled South American populations, and all of the North American populations, fall in the 'low A' half of their respective hybrid zones.

## Genetic basis of the climate barrier

To identify loci that may be contributing to a climate barrier, we looked for loci with steeper than expected ancestry clines across latitude in South America. We estimated best-fitting logistic ancestry clines at ∼542k single nucleotide polymorphisms (SNPs) across the genome by re-fitting Eq 1, where $x_i$ is the population latitude and $A_i$ is the population-mean local *scutellata* ancestry at a focal SNP, and the maximum *scutellata* ancestry $M$ is 1. Similar cline models have been fit using likelihood methods under some simplifying assumptions about the form of the errors (e.g. [66, 67]). We instead use non-linear least squares to fit cline parameters without specifying a full error model and then quantify the effects of more complex unmodeled errors (including ancestry variances and covariances) through simulation. We simulated data for 100,000 independent loci undergoing drift, which we used to estimate the expected distribution of neutral clines and calculate false-discovery rates. For each simulated locus, we independently drew a vector of population ancestry frequencies from a multivariate-normal model of drift, A ancestry ∼ MVN($\alpha$, $K$), where $\alpha$ is the vector of population mean genome-wide *scutellata* (A) ancestry proportions, and the $K$ matrix measures the expected variance and covariance in ancestry away from this mean (Fig 3), and is empirically calculated using all loci across the genome (see Methods for additional details). We limit the analysis of clines at individual loci to South America where, unlike North America, we have samples spanning both halves of the hybrid zone to inform parameter estimates. While cline analyses can be used to identify both adaptive introgression and barriers to introgression (by analysing cline center displacement in addition to cline steepness [69]), here we focus on barrier loci and approach identifying positively selected loci using alternative methods that can be applied to both hybrid zones (see "Scan for ancestry-associated selection").

We find no evidence to support a simple genetic basis or environmental threshold to the climate barrier. Ancestry clines in South America are reasonably concordant across most SNPs; 95% of cline centers fall within a 1.6 degrees latitude range, with a long tail that appears to be due to adaptively introgressing loci (identified as outliers below, see S10 Fig). We find no strongly selected individual barrier loci that exceed our 5% false-discovery-threshold for cline steepness, set by MVN simulation of background ancestry patterns. On average, individual SNP clines in South America are 960km wide (w = 8.65 degrees latitude), and the steepest cline in the genome still takes approximately 555km (w = 5 degrees latitude) to fully transition from *scutellata* to European ancestry. These wide clines, coupled with the evidence for parallel genome-wide clines in North and South America, are consistent with selection tracking smooth climate transitions over broad geographic regions rather than a discrete environmental step. Furthermore, concordance in clines across SNPs in South America suggests that many loci are associated with climate-based fitness trade-offs. Under a polygenic climate barrier, we expect locally-adapted loci to be found across the genome but steeper clines to be more commonly maintained in regions with low recombination rates. This is because selected loci create stronger barriers to gene flow when there is tight genetic linkage than when selection acts on each locus independently [70]. We test this theoretical prediction in South America and find enrichment for steeper clines in regions of the genome with low recombination. The empirical top 5% steepest clines in South America are found on all 16 chromosomes and are enriched in regions of the genome with low recombination. Steep clines comprise 12.7% CI$_{95}$[8.4%-16.5%] of loci from the lowest recombination rate quintile vs. only 3.3% CI$_{95}$[3.0%-3.6%] of loci from the highest recombination rate quintile. The average effect of recombination is a 50km decrease in mean cline width between the highest and lowest recombination rate quintiles ($\Delta b$ = 0.028, [0.017-0.038]).

## Diversity and rapid expansion

From their point of origin in Brazil, *scutellata*-European hybrid honey bees invaded much of the Americas in less than 50 years [39]. Such rapid expansion can lead to high rates of drift in the continually bottle-necked populations at the front of the wave of expansion, i.e. those populations sampled furthest from Brazil. To test this expectation, we calculated nucleotide diversity, $\pi$, for each sampled population (Fig 4). Despite much further distances traveled to the northern hybrid zone, we do not observe a more pronounced bottleneck in California than in Argentina, suggesting that the expanding wave of *scutellata*-European hybrid honey bees maintained large population sizes (and did not experience strong 'allele surfing' [71]).

*Scutellata*-European hybrid bee populations are consistently more diverse than reference bee populations because they are genetic mixtures of these diverged groups. We do, however, observe a drop in diversity in the tails of both hybrid zones starting at approximately 34.5˚ latitude from the equator. We tested whether this drop in diversity is necessarily the result of a bottleneck or can be explained solely by a cline in mean ancestry composition from more diverse *scutellata* and highly admixed genomes to less diverse European stock. To test this alternative, we predicted population diversity from a simple weighted average of A, C, and M reference allele frequencies and the observed population ancestry proportions. We find that based on ancestry composition alone, we do expect a drop in diversity across the hybrid zones, although the observed drop is slightly less than our predictions (S25 Fig).

Levels of diversity within European C and M ancestries are similar to the reference panels and stable across latitude, evidence that a diverse population of European ancestry bees hybridized with *scutellata* bees as they expanded away from Brazil. We also find high diversity within A haplotypes in both hybrid zones, again consistent with no bottleneck associated with

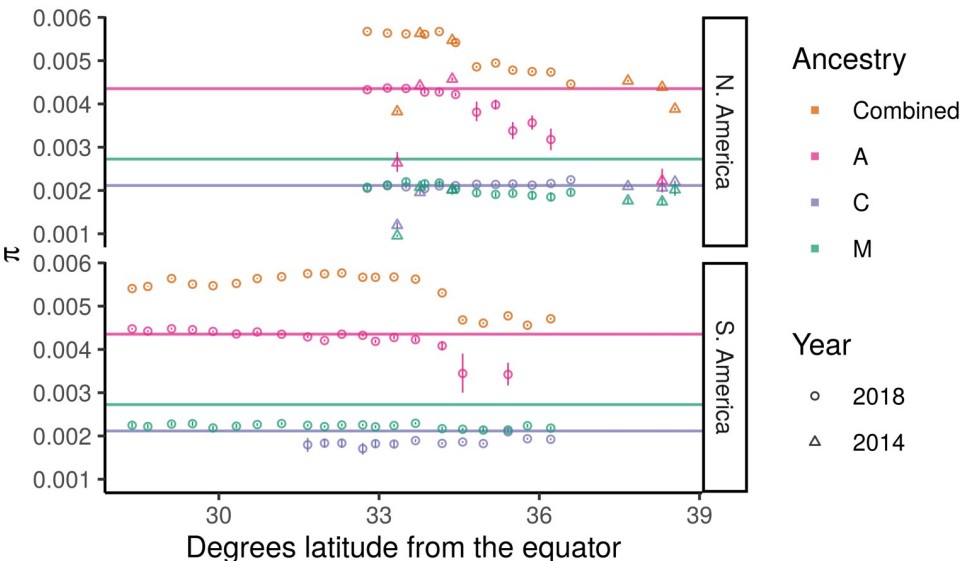

**Fig 4. Allelic diversity ($\pi$) across the hybrid zones.** For each population, we estimated allelic diversity genome-wide and within high-confidence homozygous ancestry states. Horizontal lines show the genome-wide diversity within the reference panels. Vertical lines show the 95% confidence interval for each estimate, based on a simple block bootstrap CI using 1cM blocks. For several populations in the tails of the cline, we do not show A and/or C within-ancestry estimates because these populations have too few high-confidence ancestry blocks for accurate estimation (see Methods). The low diversity outlier at 33.34 degrees latitude in the N. American cline is the 2014 Avalon sample, which comes from a small island population off the coast of California.

the rapid expansion. However, the diversity in the A ancestry background does decline in populations furthest from the equator, which is consistent with either strong filtering of *scutellata* haplotypes by selection or stochastic haplotype loss due to small *scutellata*-ancestry population sizes in the tails of the clines.

## Scan for ancestry-associated selection

We identified loci with unusually high A ancestry frequencies, a signal of natural selection, using our MVN simulations of background covariance in ancestry to set a false discovery rate. The ancestry covariances are important to account for when testing for putative selected loci that depart from genome-wide background ancestry patterns, because deviations in ancestry are correlated across populations. Although many population pairs have only small positive ancestry covariances, the cumulative effect on the tails of the distribution of A ancestry frequencies in the larger sample is striking. These covariances can confound outlier tests for selection which only consider variance from sampling (e.g. Poisson-binomial, e.g. [37]). We find that by incorporating background patterns of shared drift (or weak genome-wide selection) into our null model, we can match the bulk of the observed ancestry distributions across the genome (Fig 5).

Loci important to the successful invasion of *scutellata*-European hybrid honey bees are likely to have an excess or deficit of *scutellata* ancestry across both continents. Thus, we tested separately for high and low A ancestry outliers on each continent, and then identified overlapping outliers between the two hybrid zones. We find evidence of selection favoring *scutellata* ancestry at 0.34% of loci in N. America and 0.13% of loci in S. America, across 14 chromosomes (Fig 6A). From these outliers, we find 13 regions with an excess of A ancestry in both hybrid zones at less than a 10% false-discovery-rate (top right corner of Fig 5). The majority

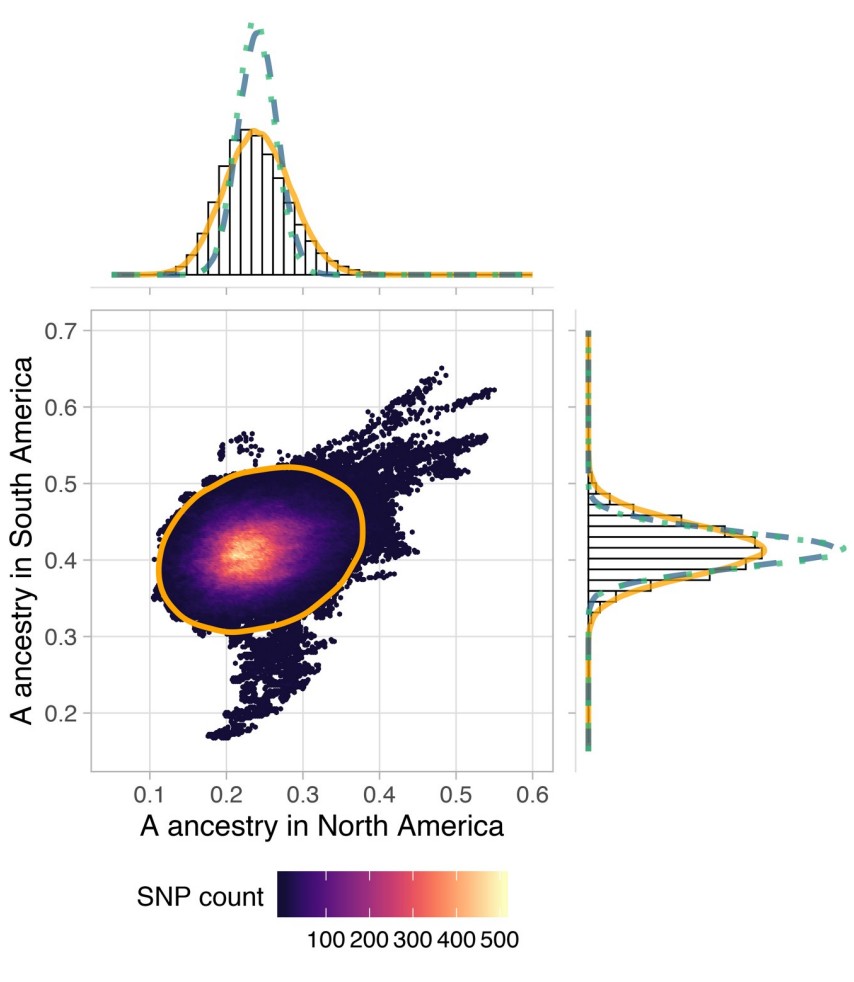

**Fig 5. Local ancestry outliers compared across hybrid zones.** We plot mean A ancestry frequency in North vs. South America for 425k SNPs across the honey bee genome. SNPs are binned for visualization, and colored by the number of SNPs within each hexagon. The orange ellipse shows the approximate 99% highest posterior density interval (HPDI) based on the full MVN model, which accounts for drift in ancestry both within and between populations. Using the same axes, we show the marginal histograms of A ancestry for each continent separately (top and right panels). Imposed on these histograms we plot density curves for 3 possible null distributions for ancestry frequencies: the full MVN model, a variance-only MVN model which only accounts for drift within populations, and a Poisson binomial model which only includes sampling variance. Most of the genome is consistent with neutrality under a MVN normal model of drift (98.6% of SNPs fall within the orange ellipse), but there are also some clear outliers. SNPs in the top right, with higher than expected A ancestry proportions in both hybrid zones, are our best candidates for loci underlying adaptive *scutellata*-ancestry associated traits. Note: While SNPs are thinned for LD, large outlier regions span many SNPs, which creates the streak-like patterns in the scatterplot.

(11/13) of shared outliers co-localize within a ∼1.5Mb region on chromosome 1, but within this region outliers separate into multiple distinct peaks (Fig 6B). One way a cluster of A ancestry peaks could form is if favored *scutellata* alleles experience additional indirect selection from being in linkage disequilibrium with other favored *scutellata* alleles at nearby loci, thereby increasing the total effective selection in a region [70]. While ancestry-informative markers (AIMs) with fixed or nearly fixed differences between *scutellata* (A) and both European (C & M) ancestries are relatively rare, we were able to confirm the highest A peak within this cluster using AIMs outside of the local ancestry inference SNP set (S20 Fig). A alleles at

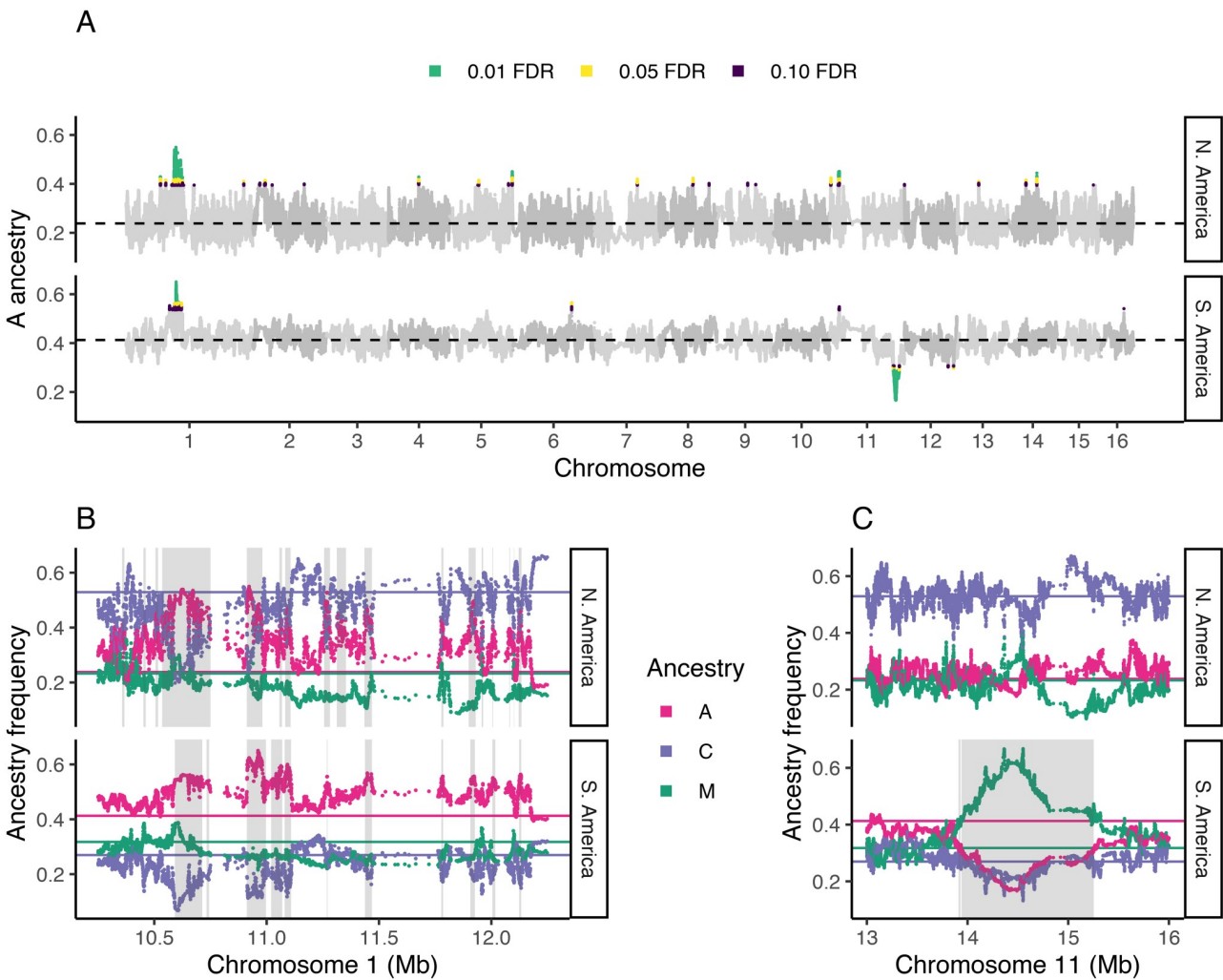

**Fig 6. Genomic location of ancestry outliers.** (A) Mean *scutellata* (A) ancestry in each hybrid zone at SNPs across the genome, with outliers colored by their false-discovery-rate. Genome-wide mean A ancestry in each zone is indicated with a dashed line. Shared peaks for high A ancestry are seen on chromosomes 1 and 11; there are no shared peaks for low A ancestry. (B) Zoomed in view of cluster of shared high A ancestry outliers on chromosome 1, with European ancestry separated into eastern (C) and western (M) subtypes. Genome-wide mean frequencies for each ancestry are shown with colored lines. Outlier regions meeting a 10% FDR for high A ancestry are shaded in grey. Shared outliers between continents overlap between the top and bottom panels. (C) Zoomed in view of the high M European ancestry outlier region found in South America. Outlier regions for low A ancestry (<10% FDR) are shaded in grey. Note: The x-axis scale differs between plots.

this main peak appear to have introgressed to high frequency hundreds of kilometers past the hybrid zone centers in both North and South America, but not reached fixation in any population (Fig 7). The rapid rise and slow fixation of A ancestry at this locus is potentially consistent with dominant fitness benefits. How far these A alleles have introgressed past the present hybrid zones is currently unknown because they exceed our range of sampling.

Our goal was to identify regions of the genome where high fitness is broadly associated with *scutellata* ancestry, but an alternative explanation for high A ancestry at a locus is that a very recent adaptive mutation just happened to fall on an A haplotype, initiating a classic 'hard sweep'. For shared high A-ancestry outlier regions, we distinguished between these two scenarios using population differentiation ($F_{ST}$) within A ancestry. We analyzed differentiation across the large cluster of shared high A-ancestry outlier peaks on chromosomes 1 and across

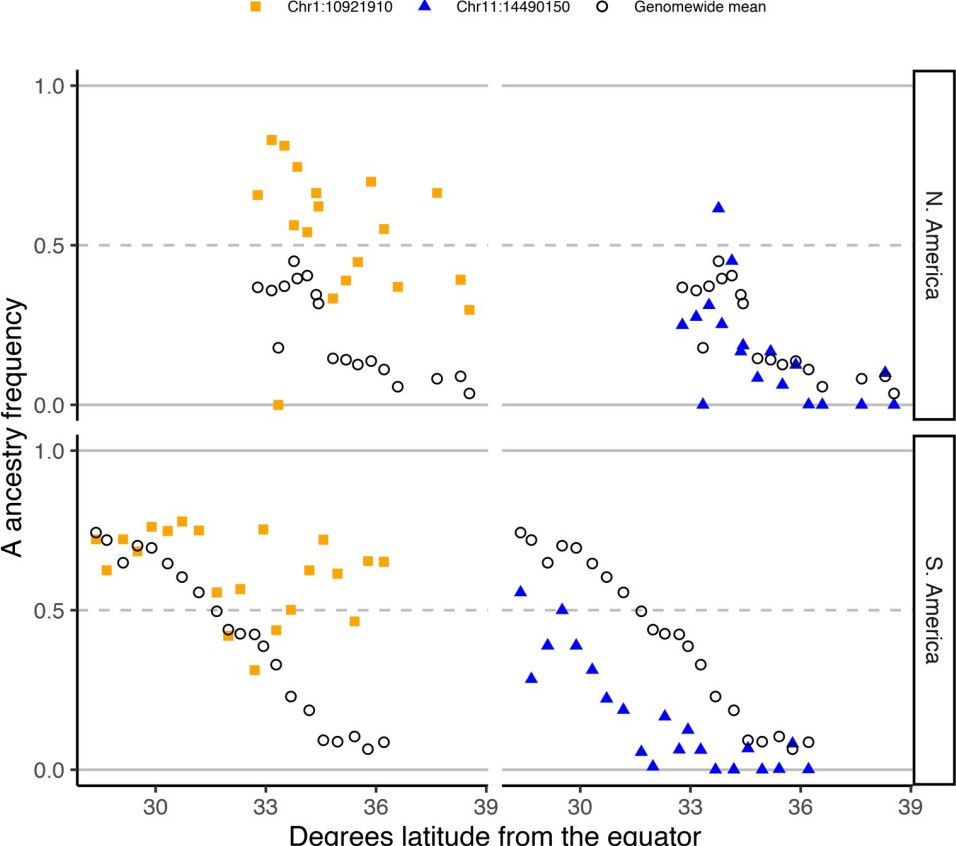

**Fig 7. Ancestry clines at outlier SNPs.** At two top outlier SNPs, we show the clines for mean population *scutellata* (A) ancestry across latitude in North (top) and South (bottom) America. To the left, we show the ancestry cline for the SNP with the highest A ancestry in the combined sample, located within the top peak on chromosome 1 for high shared A ancestry across continents. To the right, we show the ancestry cline for the SNP with the lowest A ancestry in South America, located within the large outlier region on chromosome 11 for high M and low A ancestry. Genome-wide mean local ancestry calls for each population are shown for comparison as black circles.

a smaller region on chromosome 11 that contains the other two high A ancestry outliers shared between continents. We did not find high allelic differentiation between North and South American A ancestry tracks and the *scutellata* A reference panel from Africa (S22 and S23 Figs), suggesting that *scutellata* ancestry in general, and not one particular haplotype, was favored by natural selection at these loci.

We used previous literature and gene orthologs to identify possible adaptive functions for regions of the genome where selection has favored *scutellata* ancestry. There are 3 major quantitative trait loci (QTLs) associated with defense behaviors (e.g. stinging) in genetic crosses of defensive *scutellata*-European hybrid honey bee colonies and low-defense European colonies [15, 72], none of which overlap any signatures of selection from this study. No studies have mapped the genetic basis of elevated *Varroa* defense in *scutellata*-European hybrid (vs. European) bees, but we were able to compare our results to quantitative trait loci (QTLs) associated with anti-*Varroa* hygiene behaviors [73–75] and defensive grooming [76] more generally. The cluster of peaks for high shared A ancestry on chromosome 1 overlaps a putative QTL associated with removal of *Varroa*-infested brood [75], but there are a number of large QTLs in the genome. A total of 104 genes overlap high A ancestry outlier peaks. Predicted functions for these genes (primarily based on fly orthologs) are not significantly enriched for any Gene

Ontology (GO) categories, which may simply reflect that many outlier regions are broad and contain many genes, most of which are likely unrelated to their rise in ancestry frequency. A smaller set of 37 genes with high A ancestry have signatures of selection on both continents. For these, we searched the literature and found that two have been associated with *Varroa* in previous studies: a myoneurin (LOC725494) that is overexpressed in the brains of *Varroa* infected worker bees compared to *Nosema* infected bees and controls [77] and an uncharacterized protein (LOC725683) that is over-expressed in parasitized drones compared to non-parasitized drones [78]. While these are potentially intriguing candidates for selection, *Varroa* is only one of many possible selective pressures, and more work is needed to link the signals of selection we find here to adaptive functions.

In contrast to high A ancestry outliers, we do not find any shared outliers for European ancestry. However, the most striking example of a single-zone selection event is a large 1.4Mb region on chromosome 11 with excess European ancestry in South America (bottom middle of Fig 5). This region was previously identified to have low A ancestry in *scutellata*-European hybrid honey bee populations from Brazil [37]. We independently identify this region as a low-A ancestry outlier across Argentina, using different A/C/M reference bees, ancestry calling algorithm, and bee samples than the previous paper. We find that populations across the South American hybrid zone have reduced A ancestry at this locus, but that North American populations do not appear to have experienced selection (Fig 7). By including C-lineage diversity in our admixture analysis, we additionally show that this region is specifically elevated for M haplotypes, and not European haplotypes more broadly (Fig 6C). This region has many diverged SNPs between the three ancestry groups beyond the SNPs included in our ancestry_hmm analysis, which we use to confirm high rates of M introgression (S21 Fig). It does not appear that a new mutation or narrow set of haplotypes was favored within M because we see little differentiation between the M ancestry in this selected region compared to the M reference panel. Additionally, within the selected region we find a peak of high $F_{ST}$ between A, C, and M reference panels (S24 Fig), which is consistent with this region having historically been under selection within these ancestry groups. Finally, the California bees do not have a significant deficit of A ancestry like the Argentinian bees do, but they do have two narrow peaks of excess M ancestry within this region, in the top 3% and 7% empirical percentiles for M ancestry genome-wide. Our data support a scenario in which a diversity of M haplotypes carrying the favored allele were driven to high frequency in South America after *scutellata*-European hybrid honey bees spread north of Brazil. Potential candidate genes specific to this large M-ancestry outlier region on chromosome 11 are previously described by [37]. In total, we find 186 genes that overlap low-A ancestry outlier peaks (<10% FDR). These genes are not functionally enriched for any Gene Ontology (GO) categories.

## Discussion

The introduction of *scutellata* honey bees to Brazil in the 1950s sparked one of the largest and best studied biological invasions known to date, with *scutellata*-European hybrids spreading from a single point of release over much of the Americas in less than 50 years. We add to this literature the first comparative study of the invasions in North and South America, with genome-wide resolution on the present distribution of *scutellata* (A) ancestry.

The parallel alignment of the genome-wide cline with latitude in both continents, despite very different length dispersal routes, strongly supports the view that *scutellata* ancestry has reached a stable climatic range limit. Because our transect in California only covered the upper half of the North American cline, the full shape for this genome-wide cline is uncertain, and may be asymmetrical because one signature of a moving hybrid zone is elongation of the

lagging tail [79]. In contrast, we have strong evidence for convergence in the low-A portions of these two genome-wide clines, which are not expected to be distorted by cline movement, and reflects similar latitudinal range limits for *scutellata* ancestry in North and South America. Global warming trends could shift the location of the observed clines towards the poles, as has been documented for other hybrid zones sensitive to climate change [80]. While we currently lack temporal data with comparable genomic and geographic resolution, our results can be used as a baseline for future study.

Significant effort has been focused on finding an environmental isocline that divides regions where *scutellata*-European hybrid honey bees are expected to dominate and regions where they cannot overwinter (e.g. [23–26]). However, we observe ancestry clines that are hundreds of kilometers wide, not the narrow clines created by strong selection across a discrete environmental transition. Theoretically, these broad clines could be consistent with neutral diffusion of ancestry by migration over tens of generations. However, a scenario of neutral diffusion is inconsistent with external evidence of the rapid spread of the invasion and strong fitness trade-offs with climate: the high competitive advantage and very rapid advance of *scutellata* -associated traits and A mitochondria in the tropics in the face of considerable interbreeding with European bees and, conversely, documented low fitness of *scutellata*-European hybrids in cooler climates, with low overwinter survival and maladaptive metabolic efficiency, foraging preferences and nesting behaviors (see [24] for a review). Thus, we conclude that honey bee fitness is more likely to be tracking environmental variables with smooth transitions over broad geographic regions (e.g. climate), which may create intermediate environments where ancestry intermediates have higher fitness, thus broadening the observed hybrid zones. These proposed dynamics are similar to well-studied cases in other systems where bounded hybrid superiority and/or local adaptation to continuous environments maintain adaptive clines across broad geographic regions (e.g. [81–83]).

As a null model, we expect phenotypic clines to match the scale of the observed ancestry clines, with smooth transitions in mean phenotype over hundreds of kilometers. Many phenotypes of interest, e.g. defensive behavior or *Varroa* tolerance, are expressed or measured at the colony-level and so we could not assess these in our survey of freely foraging bees. Future phenotypic surveys could be compared with our genomic clines to ascertain if key phenotypes diverge from this expected pattern, e.g. due to strong selection beyond that experienced by the rest of the genome. Indeed we see that wing length, a trait hypothesized to be associated with latitudinal body-size adaptation following Bergmann's rule [64], has a geographic distribution consistent with the genome-wide ancestry cline. This suggests that while wing length, which is strongly correlated with body size [46, 84], may well have fitness trade-offs with climate, selection for these traits does not appear to be strong enough compared to average selection for ancestry to deviate from background genomic patterns over a short time scale.

We observe relative uniformity at the climate barrier, with no individual loci showing steeper ancestry clines than what can be produced by a null model accounting for background patterns of variation in ancestry frequencies shared across populations. Nor do we observe any loci that have below 10% frequency of A ancestry in California, despite the large distance and climatic range traveled over by this portion of the invasion. If the invasive ability of *scutellata*-European hybrid honey bees were due to a small number of loci we would expect *scutellata* ancestry to have been swamped out at many unlinked neutral loci in the genome due to interbreeding at the front of the advancing wave of expansion [85]. Instead, relative genomic cohesion points to a polygenic basis for the high fitness and rapid spread of *scutellata* ancestry as well as the fitness costs in cooler climates underlying the parallel range limits observed across continents. However, we note that the distinction between so-called 'Africanized' and 'non-Africanized' honey bees is likely to further blur over time. Genetic barriers are strengthened

when selection is distributed across many loci, but they are still easily permeated by adaptive alleles [86]. Furthermore, given high recombination rates in honey bees, we predict only loci tightly associated with climate-based fitness trade-offs will remain geographically bounded over long periods of time.

Our findings add to the genomic evidence that *scutellata*-European hybrid honey bees cannot be treated as a single genetically and phenotypically cohesive group. We show that bees have intermediate *scutellata* ancestry proportions over large geographic areas, with no evidence that *scutellata*-European hybrid honey bees share any defining *scutellata* ancestry loci (including mtDNA). Colonies within these wide hybrid zones have largely unknown colony-defense behaviors and are likely to show high variance in many traits, overlapping with variation within European bees. These bees defy 'Africanized' (vs. 'non-Africanized') labels currently used by researchers, beekeepers, and policy makers. While more precise ancestry information is becoming increasingly available, it's important to understand the limitations for trait prediction. Importantly, there is no one-to-one mapping between A ancestry and colony defense. Recent findings show that both *scutellata* and M European ancestry contribute to defensiveness segregating in *scutellata*-European hybrid populations in Brazil [87]. Additionally, 'gentle Africanized honey bees' in Puerto Rico show that *scutellata*-European hybrid honey bee populations can evolve low defense while maintaining *scutellata* ancestry and other associated traits [49, 50]. Future research could improve upon ancestry-based trait predictions by identifying genetic markers for agriculturally undesirable and beneficial traits segregating in *scutellata*-European hybrid honey bee populations.

*Scutellata*-European hybrids provide a promising source of genetic variation for breeding in light of the vulnerability of European lineages to current environmental stressors and associated bee declines [88]. *Scutellata*-European hybrid honey bees have high competitive fitness and, we show here, maintained high genetic diversity despite their rapid expansion. In this study, we have taken a first step towards mapping the genetic basis of the high fitness of *scutellata*-European hybrid honey bees by identifying loci where selection has favored *scutellata* or European ancestry in both North and South America. We identify several loci with convergently high A ancestry on both continents, and many more across the genome with evidence of selection favoring A ancestry in one hybrid zone. In contrast, with the exception of one striking outlier for high M ancestry in South America, we find little evidence that European ancestry or admixture per se contributed broadly to the success of *scutellata*-European hybrid honey bees. We attribute this difference in results from a previous study of *scutellata*-European hybrid honey bees in Brazil [37] to a more appropriate null model that accounts for shared variance in ancestry across populations. While our population genetics approach is trait-blind, our results can be compared to future functional and genetic mapping studies to look for overlap between trait-associated and positively selected loci. Applying similar methods to other systems, especially where replicated hybrid zones can be sampled, holds great promise for revealing loci important to adaptation.

## Materials and methods

Statistical results and figures were created in R [57] with use of the tidyverse [58] packages. Other scripts were run using GNU parallel [89].

### Sampling

We sampled individual foraging honey bees across two hybrid zones, located at the transitions to temperate climates in North and South America. We sampled at least 10 bees each from 12 populations in California and 21 populations in Argentina (see maps, Fig 1).

For each population, we hand-netted individual foraging bees within a sampling radius of approximately 15km. Because commercial colonies are often temporarily relocated for the spring pollination season, we sampled in summer, when foraging bees are more likely to come from resident populations. We additionally included in our analyses 35 high-coverage published genomes of freely foraging bees collected from 6 populations between September 2014 and January 2015: Davis, Stebbins, Stanislaus, Avalon (Catalina Island), Placerita, and Riverside (Sky Valley and Idyllwild) [34]. While these sampled bees come from an unknown mixture of local feral and domesticated colonies, previous surveys from California have found that freely foraging bees tend to closely match feral sources, based on mtDNA composition [30]. Consistent with this view, eight of our sequenced bees from different populations in Argentina were collected close to a feral nest (< 5m), but do not appear to be ancestry outliers for their sampling locations. More specifically, we fit a general linear model (logit(A ancestry) ∼ absolute latitude + feral nest) using glm with gaussian errors in R and found no significant effect on A ancestry of sampling near a feral nest ($P = 0.97$). Based on these results and our seasonal timing, the bees in this study are likely sourced primarily from local feral populations, with some contribution from resident domesticated bee colonies.

## Lab work and sequencing

We selected a subset of 279 bees from our North and South American hybrid zones for whole genome sequencing, 8-9 bees per sampled population (see S1 Table). For each bee, we dissected wing flight muscles from the thorax and extracted DNA using QIAGEN DNeasy Blood and Tissue kits. We followed a new high-throughput low-volume DNA library preparation protocol (see [90] for details, "Nextera Low Input, Transposase Enabled protocol"). Briefly, we prepared individual Nextera whole-genome shotgun-sequencing DNA libraries using enzymatic sheering and tagmentation. Then we PCR-amplified and barcoded individual libraries using the Kapa2G Robust PCR kit and unique custom 9bp 3' indices. Finally, we pooled libraries within each lane and ran bead-based size-selection for 300-500bp target insert sizes. We targeted 4-6x coverage per bee based on a preliminary analysis of our power to replicate local ancestry calls from one of the published high coverage populations (Riverside 2014) using simulated low coverage data (S4 Fig). We multiplexed our samples across 5 Illumina HiSeq4000 lanes for paired-end 2 x 150bp sequencing. In total, we generated 5.1x mean coverage per bee for 278 samples. The 279th sample was excluded from all analyses for having extremely low (<0.1x) sequence coverage.

## Alignment and SNP set

In addition to the sequence data produced by this study, we downloaded Illumina raw read sequences for 35 previously published California genomes (PRJNA385500 [34]) and a high-quality reference panel of *A. m. scutellata* (A, n = 17), *A. m. carnica* (C, n = 9), and *A. m. mellifera* and *A. m. iberiensis* (M, n = 9) honey bee genomes (PRJNA216922 [91] and PRJNA294105 [8]) from the NCBI Short Read Archive. For all bees, we mapped raw reads to the honey bee reference genome HAv3.1 [92] using Bowtie2 very-sensitive-alignment with default parameters [93]. We then marked and removed duplicate reads with PICARD and capped base quality scores using the 'extended BAQ' option in SAMtools [94]. Using the software ANGSD [95], we identified a set of SNPs with minor allele frequency ≥ 5% in the combined sample based on read counts (-doMajorMinor 2 -doCounts 1). We excluded unplaced scaffolds (<5Mb total) and applied standard quality filters for SNP calling (base quality ≥ 20, mapping quality ≥ 30, total read depth ≤ 5500 (∼2x mean), and coverage across individuals ≥ 50%). We calculated the genetic position (cM) for each SNP using a 10kb-scale

recombination map [96] and linear interpolation in R (approxfun). We assumed constant recombination rates within windows and extrapolated positions beyond the map using the recombination rate from the nearest mapped window on that chromosome.

We identified SNPs on the mitochondria (HAv3.1 scaffold NC_001566.1) using the same pipeline as nuclear DNA above, but allowing for extra read depth (up to 100000000x). We then called consensus haploid genotypes at these SNPs for all individuals using ANGSD (-dohaplocall 2 -remove_bads 1 -minMapQ 30 -minQ 20 -doCounts 1 -minMinor 2 -maxMis 174).

## Global ancestry inference

We estimated genome-wide ancestry proportions for each bee using methods designed for low-coverage sequence data. Briefly, we combined bee genomes from the hybrid zones with reference genomes for *scutellata* (A), eastern European (C) and western European (M) bees. To reduce linkage disequilibrium (non-independence) between our markers for global ancestry inference, we thinned to every 250th SNP ($\sim$ 14k SNPs at 19kb mean spacing) before calculating genotype-likelihoods for each bee using the SAMtools method in ANGSD (-GL 1). We first ran a principal components analysis in PCAngsd [97] to confirm that genetic diversity in the hybrid zones is well-described by 3-way admixture between A, C, and M reference panels (S1 Fig). We then estimated genome-wide ancestry proportions for all bees using NGSAdmix (K = 3) [52].

## Local ancestry inference

We inferred the mosaic of *scutellata* vs. European ancestry across the genome of each bee using a hidden Markov inference method that can account for *scutellata* (A), eastern European (C) and western European (M) sources of ancestry within low-coverage *scutellata*-European hybrid honey bee genomes (ancestry_hmm v0.94 [53]). For local ancestry inference, we enriched for ancestry-informative sites by filtering for $\geq$ 0.3 frequency in one or more reference population (A, C, or M) and at least 6 individuals with data from each reference population. We subsequently thinned markers to 0.005cM spacing, because at that distance linkage disequilibrium within ancestries is expected to be low ($r^2 < 0.2$ [33]), leaving a final set of 542,655 sites for ancestry calling, or $^1/_7$ of the original SNP set. Individual bees sequenced in this study and previously published California bees have 5.42x and 14.5x mean coverage, respectively, across this final SNP set. For each population, we jointly estimated time since admixture and ancestry across the genome of each individual, using read counts from the hybrid zone and allele frequencies for A, C and M reference populations at each SNP. To generate major/minor allele counts for each reference population, we used ANGSD to call genotypes (-doPost 1) using a minor allele frequency prior (-doMaf 1) and the SAMtools genotype likelihood (-GL 1), after quality filtering (map quality $\geq$ 30, reads matching major/minor allele $\geq$ 60%, and read depth $\geq$ 6x). As additional inputs to ancestry_hmm, we used NGSAdmix results as a prior for population ancestry proportions and set the effective population size to $N_e$ = 670, 000 [37]. We modelled a simple three-way admixture scenario: starting with C ancestry, we allowed for a migration pulse from M and a second, more recent, migration pulse from A. Timing of both migration pulses were inferred from the range 2-150 generations, with priors set at 100 and 60 generations. To calculate a point estimate for each individual's ancestry proportion at a SNP, we marginalized over the posterior probabilities for homozygous and heterozygous ancestry from the ancestry_hmm output (i.e. $A = p(AA) + 1/2(p(CA) + p(MA))$.

## Ancestry covariance matrix

To explore how populations vary and covary in their *scutellata* ancestry along the genome we calculated the empirical population ancestry variance-covariance matrix ($K$), an admixture analog of a genotype coancestry matrix (e.g. [98]). The $K$ matrix is calculated using population *scutellata* (A) ancestry frequencies inferred by the local ancestry HMM, e.g. for populations i and j with mean ancestry proportions $\alpha_i$ and $\alpha_j$, and ancestry frequencies at a locus $anc_{i,l}$ and $anc_{j,l}$, their ancestry covariance calculated across all L loci genome-wide is

$$K[i,j] = \frac{1}{L}\sum_{l=1}^{L}(anc_{i,l} - \alpha_i)(anc_{j,l} - \alpha_j).$$

## Ancestry correlations between the high-latitude cline endpoints

To more formally test for excess ancestry correlations between more geographically distant (but climatically similar) populations, we grouped populations by dividing each hybrid zone into low- and high-A ancestry regions relative to the estimated latitude for the genome-wide cline center. The southernmost 11 (out of 20) of the South American populations, and all of the sampled North American populations, fall in the 'low A' half of their respective hybrid zones. We calculated mean ancestry covariances (K matrices) separately for each chromosome, using the genome-wide mean ancestry as $\alpha$, then summarised across populations by taking the mean correlation for each type of pairwise comparison, within and between continents and regions. We tested whether, on average across chromosomes, low-A South American populations share higher ancestry correlations with low-A North American populations than with geographically closer high-A South American populations and repeated this test excluding chromosomes 1 and 11 which contain large outlier regions (S14 Fig). We also tested the same group comparison across recombination rate quintiles instead of chromosomes (S15 Fig).

To investigate whether recent long-distance migration likely generated the elevated ancestry correlations we observe between low-A South America and low-A North America, we looked at patterns of allelic covariance within ancestry. Specifically, for each ancestry we estimated a genetic covariance matrix in PCAngsd for all individuals sampled from the hybrid zone, based on allelic diversity within high-confidence homozygous ancestry tracts (posterior >0.8). Under recent migration, we would expect the excess A ancestry correlations between the two ends of the hybrid zones to be mirrored by allelic covariances within all three ancestries. Instead, we find that the two most prevalent ancestries, A and C, both have low or negative genetic covariances between continents (S17 Fig). In contrast, M ancestry does show an excess of cross-continent covariance, and we followed up to determine if this is uniquely American covariance (i.e. the result of shared drift within the Americas) or could have been imported from Europe. Adding reference individuals to these within-ancestry analyses, we find that M ancestry in the Americas imported pre-existing structure from Europe, with more Poland-like (*Apis mellifera mellifera*) than Spain-like (*Apis mellifera iberiensis*) M ancestry at the temperate ends of the clines (S18 and S19 Figs).

## Simulated ancestry frequencies

At various points in the results we compare our outliers to those generated by genome-wide null models of ancestry variation along the genome. We simulated variation in ancestry frequencies at SNPs across the genome under three models: (1) A Poisson-binomial model that only accounts for sampling variance, not drift (e.g. [37]); (2) a multivariate-normal model with covariances set to zero, which accounts for effects of both sampling and drift within-

populations (e.g. [99]); and (3) a multivariate-normal model with covariances to additionally account for shared drift in ancestry between populations. For each model, we simulated in R neutral A ancestry frequencies at 100,000 independent loci [100, 101]. The full multivariate-normal model is used for comparison to the results, while the first two models are only used to show the effects of ignoring covariances.

In the Poisson-binomial simulation, for each bee we sampled 2 alleles from a binomial distribution with mean equal to the individual's genome-wide ancestry proportion inferred by ancestry_hmm.

For the variance-only MVN simulation, we empirically calibrated an independent normal distribution for each population that can exceed binomial ancestry variance (e.g. due to drift). This model is equivalent to the full MVN model below, but sets all off-diagonal entries of the $K$ ancestry variance-covariance matrix to zero.

In our full multivariate-normal model, we account for non-independent ancestry within and between our sampled populations:

$$A \text{ ancestry} \sim \text{MVN}\,(\alpha, K),$$

where $\alpha$ is the vector of genome-wide mean population ancestry frequencies and $K$ is the empirical population ancestry variance-covariance matrix. Because the MVN models are not bounded by 0 and 1, but real frequency data is, we set all simulated individual population frequencies exceeding those bounds (5.2% low and 0.09% high) to the bound. Truncation has little effect on the distribution in general and no effect on the frequency of high A ancestry outliers, but does make extremely low outliers (attributed to some populations having simulated negative frequencies) less likely (S11 Fig). For more details on model approximations to the observed data, see S12 and S13 Figs.

## Cline models

To better understand the role of dispersal and selection maintaining the current geographic range limits of *scutellata* ancestry, we fit a logistic cline model to the individual genome-wide ancestry proportions estimated by NGSAdmix. We estimated continent-specific $c$ and $b$ parameters to test for a difference in cline center (degrees latitude from the equator) and/or slope between the northern and southern invasions. Then we fit a joint model with a single cline to see how well absolute latitude or climate can consistently predict A ancestry frequencies across both continents. Specifically, we tested four environmental variables that likely contribute to varying fitness across space: mean annual temperature (˚C), mean temperature of the coldest quarter (˚C), minimum temperature of the coldest month (˚C) and mean annual precipitation (cm). We downloaded mean climate observations for 1960-1990 [68] from WorldClim.org at 30 second map resolution ($\approx 1\text{ km}^2$ at the equator) and then averaged within a 5km radius around each bee's sample coordinates. We compared climate and latitude-based selection models to a neutral dispersal model, where genome-wide A ancestry is predicted solely based on the distance (km) traveled from Río Claro, São Paulo, Brazil, the point of origin for the *scutellata*-European hybrid honey bee invasion (estimated from GPS coordinates using "distGeo" in R [102]). For each model, we substituted latitude, distance, or climate for $x_i$ in Eq 1 and we used AIC to compare model fits.

We then fit individual-SNP clines to the mean population ancestry frequencies in South America, where our samples span the full cline. We tested for individual outlier loci that may underlie a climate barrier by fitting the same logistic cline model to a set of simulated population ancestry frequencies for S. America (see MVN simulation), and comparing observed cline slopes to this null distribution. In addition, we tested for enrichment of the empirical top 5%

of steep clines in regions of the genome with low recombination rates. We divided the genome into 5 equal-sized recombination rate bins ([0, 2.92], (2.92, 21.6], (21.6, 31.7], (31.7, 38.6] and (38.6, 66.9] cM/Mb) and used 10,000 block bootstraps [103] to calculate basic bootstrap confidence intervals for each recombination rate quintile while accounting for spatial correlation in both cline slopes and recombination rates across the genome. For the bootstrap, we divided the genome into 0.2cM blocks, we re-sampled these blocks with replacement, and for each recombination bin we calculated mean $b$ and the proportion of SNPs in the top 5% steepest slopes from our bootstrap sample. When fitting non-linear least squares in R for both genome-wide and individual snp clines, we used multiple random starting values to make sure we searched across all local minima and found the global optimum solution (nls.multstart [104]). Starting values were drawn from uniform distributions: b $\sim$ Unif(-5, 5) and $\sim$ Unif(min, max) across the observed range for latitude and climate variables.

## Wing morphology

We imaged a slide-mounted fore wing and measured wing length to the end of the marginal cell using imageJ (S8 Fig). We included 269 bees in the wing analysis (only bees sequenced by this study had wings preserved and n = 9 bees were excluded for wing tatter or damage).

We also measured fore wing lengths for A, C, and M reference bees in the Oberursel Collection sampled from their native range (n = 52 [105]). While the effect of ancestry on wing length is similar in magnitude and direction in both datasets, we found that the mean wing lengths for the European reference bees (C & M) fell below the mean for our American bees with close to 100% European ancestry, perhaps reflecting phenotypic plasticity. Thus we do not incorporate these measurements of A, C, and M reference bees into the subsequent analyses.

We tested various models of the relationship between wing length, ancestry and geography. First, we fit a linear model to predict wing length in our sample from genome-wide ancestry. We visually compared our wing measurements to what we would expect if the cline in wing length across latitude were fully described by this linear relationship between ancestry and wing length and our best-fit clines for genome-wide ancestry (Fig 2). We additionally tested for differences between continents by adding a main effect and an interaction term for South America to our linear model.

We performed admixture mapping of wing length to test if the ancestry state at any individual SNP predicts residual variation in wing length. To do this, we first regressed wing length on genome-wide A ancestry, to correct for background ancestry effects. We then took the residual wing lengths from this linear model fit and regressed these on A ancestry allele counts at each locus in turn (using the maximum a posterior probability (MAP) estimates from the local ancestry HMM). We set a genome-wide significance threshold of $p < 1.1 \times 10^{-6}$ to control for multiple testing at a 5% family-wise error rate, using an analytical approximation for admixture mapping, calculated assuming 47.6 generations since admixture [106, 107].

## Identifying ancestry outlier regions and genes

To identify loci underlying ancestry-associated fitness differences, we tested SNPs for an excess or deficit A ancestry within each hybrid zone. We calculated 1%, 5% and 10% false-discovery rates (FDR) by using our MVN simulation results to set the number of false-positives we expect under a neutral model for high and low A ancestry within each continent separately (one-tailed outlier tests). We then compared the overlap in outliers between hybrid zones to identify SNPs with signatures of selection on both continents.

In addition to local ancestry, we used ancestry-informative markers with fixed or nearly fixed differences to verify high-introgression regions. We defined ancestry informative

markers as SNPs with coverage for at least 5 individuals from each reference panel and >0.95 allele frequency differences between the focal ancestry and both other ancestries. We estimated allele frequencies at each ancestry-informative marker using ANGSD, polarized SNPs so the focal ancestry has the highest MAF, and only included markers with coverage in all sampled populations. Ancestry-informative markers for A (n = 4,302) are relatively rare compared to markers for C (n = 17,384) and M (n = 15,626) because European populations each experienced a historical bottleneck differentiating them from the other two groups. Because of LD-thinning before local ancestry inference, 88% of these ancestry-informative markers were not included in the ancestry_hmm SNP set, and therefore provide separate support for high-introgression regions.

We identified a set of candidate genes that overlap regions of the genome with exceptionally high or low A ancestry (<10% FDR) using BEDtools [108]. For this analysis, we downloaded gene annotations for the HAv3.1 genome assembly from NCBI (accessed 7/22/19). 72 out of 104 genes overlapping high A ancestry peaks and 131 out of 186 genes overlapping low A ancestry peaks have associated BEEBASE gene IDs. For these high and low gene sets, we tested for enrichment of Gene Ontology (GO) terms compared to a background of all honey bee genes, using DAVID 6.8 [109] and a Benjamini-Hochberg corrected FDR of 5% [110]. To find out what is previously known about the 37 genes that overlap regions with high A ancestry on both continents, we conducted a literature search using the NCBI gene search tool and google.

We additionally checked if our candidate selected loci overlap regions of the genome previously associated with defensive or anti-*Varroa* behavioral traits (QTLs and associated marker sequences from [15, 72–76, 111–113]). We estimated genome coordinates for QTLs by blasting marker sequences to HAv3.1 and keeping the best BLASTn [114] hit with an E-value <0.01 (see S5 Table). When assessing physical overlap between genome annotations and ancestry outliers, we assumed ancestry calls for a SNP apply to the short genomic window around that SNP, spanning midway to the next ancestry call. When visualizing and counting the number of selected regions in the genome, we further merged near-adjacent (<10kb) significant ancestry windows into contiguous regions.

## Population diversity

We calculated allelic diversity ($\pi$) for each population and our A, C, and M reference panels. First, we calculated a simple unbiased population allele frequency in ANGSD based on a weighted average of observed read counts (counts -8) for each SNP. For this analysis, we included all SNPs ascertained in the combined sample (see 'Alignment and SNP set' above) but excluded SNPs from a population's estimate when fewer than two individuals had coverage. Using these allele frequency estimates, and a finite-sample size correction ($n = 2\times$ number of individuals with data at a site), we calculated mean per-SNP heterozygosity. To approximate uncertainty in our estimates, we divided the genome into 5,254 non-overlapping 1cM blocks, re-calculated our diversity estimates for 10,000 block bootstrap samples, and calculated a 95% simple bootstrap confidence interval. Finally, to get per-bp diversity, we scaled our per-SNP diversity estimates by the density of SNPs in the genome, using the same coverage and depth quality filters in ANGSD as in our SNP pipeline to count total mappable sites.

For within-ancestry diversity estimates, we used our ancestry calls to identify contiguous tracts with high posterior probabilities (>0.8) of homozygous A, C, or M ancestry. We used these tracts to divide the genome into high confidence A, C, and M ancestry states, and filter for reads that mapped within these states. We then repeated the estimation and block bootstrap procedure above using only the reads associated with a particular ancestry. To estimate within-ancestry diversity for a population, we required data for at least 75 1cM blocks spread

across at least 15 of the 16 chromosomes, which excludes 6 populations with too little A ancestry in the tails of both clines and 8 populations with too little C ancestry in the S. American cline for accurate estimation. We compared observed and predicted heterozygosity for each population based on expected allele frequencies calculated by multiplying population-specific admixture proportions by reference population allele frequencies for each ancestry.

To test whether selection had favored specific haplotypes, or *scutellata* ancestry more generally, within shared outlier regions for high A ancestry, we calculated population differentiation ($F_{ST}$) between the A reference panel and A haplotypes within each hybrid zone. We also calculated within-ancestry $F_{ST}$ between the two hybrid zones, to assess whether the same A haplotypes rose in frequency on both continents. Likewise, for the large high M outlier on chromosome 11, we calculated pairwise differentiation within M ancestry between North America, South America, and the M reference panel. We similarly calculated $F_{ST}$ for all three contrasts between A, C, and M, reference panels across these outlier regions, to test for signatures of historical selection and divergence between these ancestry groups. For $F_{ST}$ calculations, we estimated within-ancestry allele frequencies for North and South America using the same method described above for within-ancestry $\pi$, except pooling individuals by hybrid zone rather than population. We used Hudson's estimator for $F_{ST}$ (Eq 10 in [115]), calculated the average per-SNP $F_{ST}$ within sliding 50kb windows stepping every 1kb across ancestry outlier regions, and only included SNPs with coverage for at least two individuals for both populations in the contrast and windows with at least 10 SNPs.

### Ethics statement

Honey bee samples from California were collected with permission from the California Fish and Wildlife (wildlife.ca.gov; permit ID D-0023599526-1). Honey bee samples from Argentina were collected and transferred to the University of California, Davis, for genomic analysis with authorization from Argentina's National Institute of Agricultural Technology (Instituto Nacional de Tecnología Agropecuaria, INTA Argentina, www.argentina.gob.ar/inta, document ID 25401; MTA No. 2018-0374-M filed at UC Davis). This study did not involve any endangered species, protected species or protected areas.

## Supporting information

**S1 Table. Sample information.** Geographic sampling information (population, location, date, whether collected by a feral nest), approximate sequencing coverage, global ancestry estimates and wing lengths for bees sequenced in this study and reference bees.
(TXT)

**S2 Table. Cline model comparison.** Model rankings between logistic cline fits for genome-wide *scutellata* (A) ancestry predicted by climate and distance variables.
(PDF)

**S3 Table. Ancestry outlier regions.** Genome coordinates for outlier regions with high or low *scutellata* (A) ancestry. Adjacent and near adjacent (within 10kb) ancestry windows meeting <10% FDR have been combined into contiguous regions and are labelled with the lowest FDR within the region. Note that shared high A outlier regions, by definition, will overlap high A South American and high A North American outlier regions, with bp and percent overlap listed. NA signifies not significant for that hybrid zone.
(TXT)

**S4 Table. Ancestry outlier genes.** List of genes overlapping ancestry outliers at 1%, 5%, and 10% FDR thresholds. Minimum FDR for each continent listed separately. NA signifies not

significant for that hybrid zone.
(TXT)

**S5 Table. Approximate QTL coordinates HAv3.1.** Approximate coordinates (HAv3.1) for regions of the genome previously associated with defensive behaviors or *Varroa* tolerance.
(TXT)

**S6 Table. Invasion dates and locations.** Approximate locations and dates of first arrival for the spread of *scutellata*-European hybrid honey bees as plotted in Fig 1. We estimated GPS coordinates for each historical observation using google maps and the available location description.
(TXT)

**S1 Fig. PCA.** Principal components analysis generated in PCAngsd using genotype likelihoods from the same thinned set of 14,044 autosomal SNPs used in global admixture analysis. The major axes of diversity separate out C ancestry (PC1) and M ancestry (PC2). Consistent with 3-way admixture, all sampled bees from North and South America are intermediate on the PCA, in the triangle formed by reference panels for *Apis mellifera scutellata* from southern and eastern Africa (A), *A. m. carnica* from eastern Europe (C) and *A. m. mellifera* and *A. m. iberiensis* from western Europe (M).
(TIF)

**S2 Fig. Ancestry in reference panels.** Results of NGSAdmix global admixture analysis for reference populations from the combined analysis of all populations (K = 3). These results were used to assign the unlabelled ancestry components output by NGSAdmix to A, C, and M groups, based on a clear mapping to the three reference populations. We see a small amount of admixture between C and M within our reference populations, which is consistent with limited gene flow from secondary contact within Europe.
(TIF)

**S3 Fig. Comparison of local and global ancestry results.** (A) Comparison of the mean genome-wide ancestry estimate from NGSAdmix (x-axis) and ancestry_hmm (y-axis) for each bee, with one-to-one line drawn in grey. The mean for the HMM is calculated by marginalizing the posterior over all ancestry states and taking a mean across SNPs. The individual-level ancestry estimates between the two methods agree strongly (Pearson's correlation: 0.997 A, 0.999 C, 0.985 M), but the HMM estimates slightly higher minor ancestry for bees with low admixture proportions. (B) Population mean summarises for the same comparison of NGSAdmix vs. ancestry_hmm genome-wide ancestry estimates, with one-to-one line drawn in gray. Because the population mean ancestry proportions from NGSAdmix are used as a prior for the population-specific mixing proportions in ancestry_hmm, this panel can also be interpreted as the prior (x-axis) and posterior (y-axis) of the local ancestry HMM for population-level admixture proportions.
(TIF)

**S4 Fig. Power to call local ancestry.** (A) Correlation between high-coverage and low-coverage ancestry calls, across different simulated depths of coverage (1-10x). (B) Proportion of high-confidence ancestry calls from high-coverage data that were replicated in analyses of low-coverage data, with different simulated depths of coverage (1-10x). These results are from a preliminary analysis of the power to call local ancestry accurately, used to inform target sequencing depth for this study. For this preliminary study, we used a published SNP set with data for A, C, and M reference populations [8] based on earlier versions of the honey bee genome (Amel4.5 [116]) and recombination map [33]. We enriched for ancestry-

informativeness and thinned for linkage disequilibrium ($\geq 0.2$ MAF in at least one reference population and $r^2 < 0.4$ within the A reference population), leaving 161k SNPs. First we ran ancestry_hmm [53] using called genotypes from a high-coverage admixed population with intermediate admixture proportions (Riverside 2014 (n = 8): 40% C, 20% M, 40% A ancestry). We simulated lower coverage data from this same population by generating a binomial sample of $n$ reads for each locus, based on the individual's genotype. To simulate realistic variance in coverage across the genome, $n$ for each site and individual was generated from a negative binomial distribution with variance 3x the mean [117]. We additionally simulated a 1% sequencing error rate. Running local ancestry inference on the high coverage data, we inferred high confidence ancestry states for 81% of sites. First we calculated a point estimates for A ancestry ($p(AA) + ^1/_2(p(CA) + p(MA))$) at every site for each individual and used these estimates to calculate a correlation between the high coverage ancestry calls and low coverage ancestry calls. Then we calculated the percent of high confidence calls that were replicated with high confidence ($>0.8$ posterior) in the low coverage data for the same ancestry state ("correct") or a different ancestry state ("incorrect"). Call to the HMM for simulated low-coverage data: ancestry_hmm -e 3e-3 -a 3 0.4 0.2 0.4 -p 0 100000 0.4 -p 1 -100 0.2 -p 2 -60 0.4 –tmax 150 –tmin 2 –ne 670000. For original high coverage data we used genotype calls rather than read counts (-g) and a lower error rate (-e 1e-3).
(TIF)

**S5 Fig. Estimated generations post-admixture.** Inferred timing of migration pulses from A ancestry (left) and M ancestry (right). Each population's admixture timing is estimated separately, during local ancestry inference (ancestry_hmm), and results are plotted across latitude. We allowed a range of 2-150 generations, so the highest time estimates are truncated at 150 generations. Admixture with *scutellata* (A) ancestry began in 1956, 62 years before sampling in 2018. We have little prior information about the timing of M into C admixture, which likely varies across the Americas, but in general should pre-date admixture with A. The number of generations per year for feral honey bee populations is uncertain.
(TIF)

**S6 Fig. Climate variables across latitude.** Bioclim climate variables for all sample sites plotted against latitude: (A) Mean annual temperature (B) Mean annual precipitation (C) Mean temperature coldest quarter (D) Minimum temperature coldest month. Two adjacent climate outliers in the N. American sample can be seen in the top two panels and represent bees from an inland desert (hot and dry) and a high altitude sampling site (cold and wet) at similar latitudes in Riverside County, CA. Bees from this same high altitude site are also outliers in the bottom two panels, having the coldest mean and minimum winter temperatures of all sites.
(TIF)

**S7 Fig. Wing length predicted by ancestry.** Individual honey bees are represented as points, with wing lengths plotted along the x-axis and genomewide A ancestry proportions (NGSAdmix results) along the y-axis. We draw the best-fit regression line (slope = -0.72 mm, $F(1, 267)$ = 119, $P = 3.65 \times 10^{-23}$, $R^2 = 0.31$, n = 269). We also include wing lengths for A, C and M reference bees from the Oberursel Collection, which we assume have none or full A ancestry. These reference bees are plotted slightly outside the range [0, 1] and with jitter to facilitate viewing individual points that would otherwise all cluster on the boundaries.
(TIF)

**S8 Fig. Wing length measurement.** Fore wing image cropped and annotated to show length measurement taken. A full length to the tip of the wing is the standard measurement, but we

use this alternative because many of our samples have significant wing tatter.
(TIF)

**S9 Fig. Admixture mapping analysis.** We plot the p-value for each SNP across the genome, based on independent tests of association between A ancestry at that SNP and wing length. The red dashed line marks the genome-wide significance threshold for a family-wise error rate of 0.05, using a two-tailed test. In admixture mapping, SNPs are correlated, and the number of independent statistical tests depends on the number of generations recombination has had to break up ancestry blocks. Here we use an analytical approximation for the significance threshold based on 47.65 generations of admixture (population median estimate).
(TIF)

**S10 Fig. Distribution of ancestry clines in South America across SNPs.** A logistic cline model was fit to observed and simulated population ancestry frequencies across latitude for S. America. Estimated cline parameters, center and width ($w = |^4/_b|$), are presented as violin plots. Units for both cline center and width are degrees latitude. We additionally partition observed SNPs by outlier and non-outlier status, set by 10% FDR for high or low A ancestry in South America. Individual SNP clines were only fit in South America, where we observed the full cline.
(TIF)

**S11 Fig. Effect of truncating MVN simulated ancestry frequencies.** (A) Percent of simulated population A-ancestry frequencies exceeding lower (left) and upper (right) bounds, and thus truncated to [0, 1] range. Each population is a point and the populations most affected by truncation are low-A ancestry populations with mean A-ancestry proportions close to the bound at 0. (B) QQ-plot comparing the quantiles for mean A ancestry before and after truncation in N. America (left) and S. America (right). The distribution of mean A ancestry is mostly unaffected by restricting simulated population A ancestry frequencies to the [0, 1] range, but truncation does reduce model predictions of very low A-ancestry frequencies in N. America, where mean A ancestry is already low.
(TIF)

**S12 Fig. Simulated vs. observed A ancestry quantiles.** QQ-plots comparing observed quantiles for mean A ancestry in North America (top) and South America (bottom) to the quantiles generated by three simulated distributions (left-to-right): MVN, MVN with zero covariances, and Poisson binomial model. Only the MVN model, allowing for covariances between populations, matches the bulk of the observed distribution.
(TIF)

**S13 Fig. Overlay of 2D density plot for observed A ancestry frequencies in North and South America compared to simulated A ancestry frequencies under a multivariate-normal model.**
(TIF)

**S14 Fig. Mean ancestry correlation by chromosome across populations.** Mean ancestry covariances (K matrices) were calculated separately for each chromosome, using the genome-wide mean ancestry as $\alpha$, then correlations were summarised by taking the mean for each type of population comparison, within and between continents and low vs. high A regions. Error bars show the normal-approximated 95% confidence intervals around these means. We divided populations in South America into two groups, 'low A' and 'high A', relative to the cline center. Low A populations are found at higher latitudes and correspondingly cooler climates. All North American samples come from the low-A side of the cline. On average across

chromosomes, low-A South American populations share higher ancestry correlations with low-A North American populations than with geographically closer high-A South American populations (0.032: $CI_{95}$[0.011,.053], P = .005, paired 2-sided t-test). Two chromosomes harbor large outlier regions consistent with their elevated correlations shown here: Chromosome 1 has a large cluster of loci with high A ancestry in North and South America while chromosome 11 has a wide region of low A ancestry exclusive to South America. The results do not change qualitatively if these two outlier chromosomes are both removed from the analysis (0.035: $CI_{95}$[0.022,0.047], P = $4.8x10^{-5}$, paired 2-sided t-test).
(TIF)

**S15 Fig. Mean correlation for population pairs by recombination rate.** Mean ancestry covariances (K matrices) were calculated separately for each of the 5 recombination rate quintiles, using the genome-wide mean ancestry as $\alpha$, then correlations were summarised by taking the mean for each type of population comparison, within and between continents and low vs. high A regions. About half of the South American populations, and all of the sampled North American populations come from the low-A side of the hybrid zone (relative to the estimated cline center). The genomewide mean is additionally shown as an X. On average across recombination bins, low-A South American populations share higher ancestry correlations with low-A North American populations than with geographically closer high-A South American populations (0.049: $CI_{95}$[0.021,.078], P = .009, paired 2-sided t-test).
(TIF)

**S16 Fig. Ancestry covariances across populations.** Ancestry covariance matrix (see Methods). Populations are ordered by latitude, with high and low A sides of each hybrid zone defined relative to the estimated genomewide cline center. Within-population variances (left) are shown separately from between-population covariances (right) because of the drastically different scales. Populations near the cline center have higher ancestry variances (most clearly seen in the diagonal elements) because they have A ancestry proportions closer to 50%. Drift and finite sample sizes also contribute to the observed ancestry variances. The third lowest latitude population in the North American cline with exceptionally high ancestry variance is Avalon, sampled from Catalina Island off the coast of California. The observed excess covariance between the two distant ends of these clines is unexpected under a simple model of spread North and South out of Brazil.
(TIF)

**S17 Fig. Genetic covariance within ancestry.** Genetic covariances within A (top), C (middle) and M (bottom) ancestry. Colors represent the population mean genetic covariance between individuals, and the range of values varies by ancestry (note: color bars have different scales). While kinship creates strictly positive covariances, here we observe some negative values because we can only calculate co-variation around the empirical mean combined sample allele frequency, not the true ancestral allele frequency (which is unknown). Population mean covariances were summarised from an individual-by-individual covariance matrix generated using PCAngsd from bam files filtered to only include regions of the genome with high confidence homozygous ancestry calls for the focal ancestry (posterior >0.8 from ancestry_hmm). High and low A sides of each hybrid zone are defined relative to the estimated genomewide cline center.
(TIF)

**S18 Fig. Principle components analysis of genetic variation within ancestry.** PCA analysis of A (top), C (middle) and M (bottom) ancestry. Analysis was performed using PCAngsd using all reference samples of the focal ancestry and sequence data from the hybrid zones

filtered to only include regions of the genome with high confidence homozygous ancestry calls for the focal ancestry (posterior >0.8 from ancestry_hmm). Each bee is a point, colored by sample location. The two hybrid zones form somewhat separable clusters for European (C and M) ancestry, but not *scutellata* (A) ancestry. The major axis of genetic diversity within M ancestry in the Americas (PC1) mirrors pre-existing population structure within Europe between *Apis mellifera mellifera* (Poland) and *Apis mellifera iberiensis* (Spain), two well-known honeybee subspecies that may have different historical import rates to different regions.
(TIF)

**S19 Fig. Genetic variation within ancestry by latitude.** Here we plot the first principal component for genetic diversity within A (top), C (middle) and M (bottom) ancestry against absolute latitude of sampling location within the hybrid zone (see S18 Fig for original PCA). Each bee is a point and reference bees are plotted to the side (not at their actual latitude). Bees are colored by sample location. Note that bees with very low amounts of the focal ancestry (higher latitudes A ancestry or lower latitudes C ancestry) fall close to zero on PC1 (dashed line), which, despite using a method designed to account for low coverage data (PCAngsd), may simply be an artifact of low information per individual bee for genetic diversity within a low-frequency ancestry. A ancestry shows very little population structure along PC1 by continent or latitude. C is the dominant ancestry at higher latitudes in both zones and shows greater separation between the two ends of the zones (higher absolute latitude for both) than within South America. M ancestry at lower latitudes in South America is more similar to *Apis mellifera iberiensis* (Spain) than M ancestry elsewhere in the Americas.
(TIF)

**S20 Fig. Ancestry and AIM frequencies for high shared A outliers on chr1.** Zoomed-in view of the region on chromosome 1 with a cluster of high A ancestry peaks in both North America (left) and South America (right), with shared outlier regions meeting a 10% FDR for high A ancestry on both continents shaded in grey. (Top) *Scutellata* (A), western European (M) and eastern European (C) local ancestry estimates at each HMM marker. (Bottom) Mean frequency of ancestry informative markers AIMs (see Methods), most of which were not included in the ancestry_hmm inference ('AIM only') due to thinning.
(TIF)

**S21 Fig. Ancestry and AIM frequencies for low A outlier region on chr11.** Zoomed-in view of the 1.4Mb region on chromosome 11 with high western European ancestry (M) in South America (right) but not in North America (left), with the outlier region meeting 10% FDR highlighted in grey. (Top) *Scutellata* (A), western European (M) and eastern European (C) local ancestry estimates at each HMM marker. (Bottom) Mean frequency of ancestry informative markers AIMs (see Methods), most of which were not included in the ancestry_hmm inference ('AIM only') due to thinning.
(TIF)

**S22 Fig. Differentiation across shared high A outliers on chr1.** $F_{ST}$ across the region on chromosome 1 with shared high A ancestry outliers. Outlier regions meeting a 10% FDR in both hybrid zones are highlighted in darker grey, while those meeting a 10% FDR in only one hybrid zone are highlighted in lighter grey. Per-SNP $F_{ST}$ is averaged within sliding 50kb windows. In addition to the three ancestry reference panels (A, C, & M), we include contrasts for the subset of individuals in each hybrid zone with high-confidence homozygous A ancestry.
(TIF)

**S23 Fig. Differentiation across shared high A outliers on chr11.** $F_{ST}$ across the region on chromosome 11 with shared high A ancestry outliers. Outlier regions meeting a 10% FDR in both hybrid zones are highlighted in darker grey, while those meeting a 10% FDR in only one hybrid zone are highlighted in lighter grey. Per-SNP $F_{ST}$ is averaged within sliding 50kb windows. In addition to the three ancestry reference panels (A, C, & M), we include contrasts for the subset of individuals in each hybrid zone with high-confidence homozygous A ancestry. (TIF)

**S24 Fig. Differentiation across low A outlier region on chr11.** $F_{ST}$ across the 1.4Mb region on chromosome 11 with high western European ancestry (M) in South America but not in North America (left), with the outlier region meeting 10% FDR highlighted in grey. Per-SNP $F_{ST}$ is averaged within sliding 50kb windows. In addition to the three ancestry reference panels (A, C, & M), we include contrasts for the subset of individuals in each hybrid zone with high-confidence homozygous M ancestry. Windows are dropped if fewer than 10 SNPs have 2 individuals with data, which produces gaps in the contrasts with N. American M ancestry because this hybrid zone does not have elevated M ancestry in this region and at many SNPs very few individuals have high-confidence homozygous M ancestry. Top peaks in M vs. C, M vs. A, and C vs. A contrasts seen within this region reach the 99.5, 98.0, and 99.8 percentiles (respectively) for 50kb windows genome-wide. (TIF)

**S25 Fig. Comparison of observed and predicted diversity.** (A) Observed and predicted allelic diversity ($\pi$) for each population across latitude. To predict $\pi$ for a specific population, we calculated the expected allele frequency based on a mixture of A, C, and M reference population allele frequencies, weighted by the population's estimated admixture fractions of these three ancestries. (B) Plot of predicted vs. observed $\pi$ for each population for direct comparison to the 1-to-1 line (dashed). Avalon (California 2014, marked as a triangle) is a clear outlier, with low diversity for its admixture fraction. (TIF)

**S26 Fig. Mitochrondrial clines.** Out of 82 SNPs on the mitochondria, we identified two with more than 80% estimated frequency difference between *scutellata* (A) and European (C & M) reference panels. Estimated allele frequencies at these SNPs for each population in North America (left) and South America (right) are plotted in color. For comparison, population mean genomewide A ancestry proportions (NGSAdmix) are plotted as open black circles. At both SNPs, estimated M and C allele frequencies are zero (not shown) and estimated A allele frequencies are high but not at fixation (plotted as pink dashed lines). Bees sequenced in this study have low coverage across most of the mtDNA sequence, which prevented us from constructing a phylogenetic tree for full mitochondrial haplotypes and creates uncertainty in our allele frequency estimates here. To reflect this uncertainty, points are shaded by the sample size (i.e. the number of mtDNA haplotypes in a population for which we were able to call a consensus base). (TIF)

## Acknowledgments

We thank the National Institute of Agricultural Technology (INTA), Argentina, for the use of bee samples from Argentina. Thank you to Stefan Fuchs and the Oberursel Collection for sharing wing images of A, C and M reference bees. Thank you to Brock Harpur for making marker sequences from previous Hunt lab QTL studies publicly available on Data Dryad. Thank you

to Julie Cridland and Nicholas Saleh for early tips on sampling and California bees, to Philipp Brand, Brenda Cameron, and Anne Lorant for sharing protocols and advice on lab work, to Jodie Jacobs and Petra Silverman for help phenotyping wing traits, and to Kelsey Lyberger, Roisin McMullen and Jodie Jacobs for their help in the field netting bees. Thank you to Jennifer Van Wyk, Hannah Whitehead, and Ang Roell for useful conversations about how scientists and the public describe *scutellata*-European hybrid honey bees and the broader impacts. Thank you to Daniela Zarate, Jeffrey Ross-Ibarra, Michael Turelli, Peter Ralph and to members of the Coop, Ramírez, and Ralph labs for insightful discussions and feedback on earlier drafts of the manuscript.

## Author Contributions

**Conceptualization:** Erin Calfee.

**Formal analysis:** Erin Calfee.

**Funding acquisition:** Erin Calfee, Santiago R. Ramírez, Graham Coop.

**Investigation:** Erin Calfee, Marcelo Nicolás Agra.

**Methodology:** Erin Calfee, Graham Coop.

**Resources:** María Alejandra Palacio, Santiago R. Ramírez, Graham Coop.

**Software:** Erin Calfee.

**Visualization:** Erin Calfee.

**Writing – original draft:** Erin Calfee, Graham Coop.

**Writing – review & editing:** Erin Calfee, Marcelo Nicolás Agra, María Alejandra Palacio, Santiago R. Ramírez, Graham Coop.

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
