## [Decision Letter · Decision Letter 0]

18 May 2020

Dear Dr Calfee,

Thank you very much for submitting your Research Article entitled 'Selection and hybridization shaped the Africanized honey bee invasion of the Americas' to PLOS Genetics. Your manuscript was fully evaluated at the editorial level and by independent peer reviewers. The reviewers appreciated the attention to an important topic but identified some aspects of the manuscript that should be improved.

We therefore ask you to modify the manuscript according to the review recommendations before we can consider your manuscript for acceptance. Your revisions should address the specific points made by each reviewer.

[LINK]

Yours sincerely,

Alex Buerkle

Associate Editor

PLOS Genetics

Kirsten Bomblies

Section Editor: Evolution

PLOS Genetics

This manuscript has been carefully reviewed by two referees, each of whom is enthusiastic about the manuscript and its publication. The reviews include only a few questions and requests for clarification, which I ask the authors to consider in preparing a revised manuscript. I appreciate the care the authors have taken in preparing a very clear manuscript to describe remarkable research.

Reviewer's Responses to Questions

**Comments to the Authors:**

Reviewer #1: In this MS, Calfee and colleagues present a well-crafted and careful study of the genomics of Africanized honey bee invasion in the Americas. The authors demonstrate quite convincingly that the invasion has likely reached some climactic limit, and that adaptive traits of the invasive bees are broadly polygenic. The authors find several candidate regions in which ancestry proportions and clines deviate significantly from the remainder of the genome, and give plausible selection scenarios to explain these regions. All in all I found the paper anticipated potential issues for most results very well, and does a great job of conservatively interpreting the results of its many analyses.

Major Comments:

I think the only place where I desired a bit more of an in-depth look was in looking at ancestry correlations at the extreme ends of the clines (Figures 3, S14,S15). The authors present one potential explanation in the form of convergent selection, and suggest the pattern is not driven by any particular region, but Figure S14 seems to show a distribution of Cold NA vs Cold SA driven largely by chromosome 1. Additionally, it's not clear to me how convergent selection could drive ancestry correlations to be higher between NA and SA samples than they are within the cold SA samples. One potential driver I could think of would be recent shared ancestry between honey bees from both NA and SA, perhaps due to human trading. I would love to a see a genome-wide gene tree/network for all of the samples to see if there is some evidence for some clustering there as well. In general, I think a gene tree of your samples, while not essential, would be helpful in aiding readers interpret some of the results.

Minor Comments:

"Cold vs Warm" terminology: I found this terminology both confusing and not very useful as the two parts of the cline are determined by ancestry and not by climate. I understand that there *are* broad climactic differences as well, but perhaps "High A" vs "Low A" or similar could be used, as I found myself thinking purely in terms of climate and not ancestry.

Line 140: you should define b, not w, here, especially since you don't seem to use w in the rest of the main text.

Figure 7: Add label for the genome wide patterns similar to the two outlier SNP categories.

Line 428: Purely stylistic - I thought I had missed some big results about wing length with that introduction. I completely agree that admixture mapping is potentially extremely powerful in this system, but perhaps just say that as you have not found any significant results to serve as an example.

Reviewer #2: Review of “Selection and hybridization shaped the Africanized honey bee invasion of the Americas”

In this paper, the authors explore the spread of African honey bee ancestry in North and South America. They sampled and sequenced populations along transects spanning dual hybrid zones at either end of Africanized bee ancestry. The authors find that ancestry and wing size clines are surprisingly consistent in both continents, although they also highlight several ancestry outlier regions, suggesting the action of selection.

This is a very, very good paper. I’ve never actually done this before, but I don’t have any suggestions or critiques. This paper succeeds at two levels. On the topic of africanized honey bees, the finding that the hybrid zone is consistent between continents is really important for understanding what limits the spread of africanized ancestry. The authors also find several regions that look like they could be under selection for different ancestry groups, which are intriguing now and could turn up important features of bee biology if probed further. At an evolutionary biology level, this paper is a great example of how selection can create the bounds of a hybrid zone. The authors also do an amazing job of just being very careful and thorough. I really appreciate the effort to simulate realistic null distributions. Altogether, this paper is a great example of how to do a population genomics analysis. I commend the authors.

Minor note:

Line 545: “…bees sequenced in y…” is that correct?

**Have all data underlying the figures and results presented in the manuscript been provided?**

Reviewer #1: Yes

Reviewer #2: Yes

PLOS authors have the option to publish the peer review history of their article (what does this mean?). If published, this will include your full peer review and any attached files.

Reviewer #1: No

Reviewer #2: No

---

## [Editor Report · Decision Letter 1]

9 Aug 2020

Dear Dr Calfee,

We are pleased to inform you that your manuscript entitled "Selection and hybridization shaped the rapid spread of African honey bee ancestry in the Americas" has been editorially accepted for publication in PLOS Genetics. Congratulations!

Yours sincerely,

Alex Buerkle

Associate Editor

PLOS Genetics

Kirsten Bomblies

Section Editor: Evolution

PLOS Genetics

Comments from the reviewers (if applicable):

This revised manuscript and accompanying letter respond very clearly and completely to the suggestions for improvement of the manuscript. These include responses to comments from the previous reviewers and from other readers. I appreciate the authors' attention to many details and their clear description of the changes that have been made.

**Data Deposition**

http://datadryad.org/submit?journalID=pgenetics&manu=PGENETICS-D-20-00578R1

**Press Queries**

---

## [Editor Report · Acceptance letter]

12 Oct 2020

PGENETICS-D-20-00578R1 

Selection and hybridization shaped the rapid spread of African honey bee ancestry in the Americas 

Dear Dr Calfee, 

We are pleased to inform you that your manuscript entitled "Selection and hybridization shaped the rapid spread of African honey bee ancestry in the Americas" has been formally accepted for publication in PLOS Genetics! Your manuscript is now with our production department and you will be notified of the publication date in due course.

With kind regards,

Matt Lyles

PLOS Genetics

On behalf of:
